# Effect of Street Asymmetry, Albedo, and Shading on Pedestrian Outdoor Thermal Comfort in Hot Desert Climates

Hakima Necira [1], Mohamed Elhadi Matallah [1,2], Soumia Bouzaher [1], Waqas Ahmed Mahar [2,3,*] and Atef Ahriz [4]

1  Laboratory of Design and Modelling of Architectural and Urban Forms and Ambiances (LACOMOFA), Department of Architecture, University of Biskra, Biskra 07000, Algeria; hakima.necira@univ-biskra.dz (H.N.); elhadi.matallah@univ-biskra.dz (M.E.M.); s.bouzaherlalouani@univ-biskra.dz (S.B.)
2  Sustainable Building Design (SBD) Lab, Department of Urban and Environmental Engineering (UEE), Faculty of Applied Sciences, Université de Liège, 4000 Liège, Belgium
3  Department of Architecture, Faculty of Architecture and Town Planning (FATP), Aror University of Art, Architecture, Design and Heritage, Sukkur 65200, Pakistan
4  Laboratory of Applied Civil Engineering (LGCA), Echahid Cheikh Larbi Tebessi University, Tebessa 12022, Algeria; atahriz@gmail.com
*  Correspondence: architectwaqas@hotmail.com

**Abstract:** Improving urban walkability in the face of climate change is a critical challenge for urban designers. Street design strategies can mitigate heat stress and enhance pedestrian livability. Most previous studies conducted in hot climates recommend adopting deep canyons to improve summer conditions, overlooking the potential improvement of wide streets as essential structural elements of the urban fabric. This study was conducted in Biskra city, southern Algeria, where several mitigation strategies were applied to 'Emir Abdelkader Boulevard', as the main structural street inside the city, to create an optimal street model for arid climates. Five scenarios were developed based on three criteria: (Sc1) asymmetric profile northeast side (NES) > southwest side (SWS); (Sc2) asymmetric profile SWS > NES; (Sc3) cool paving; (Sc4) horizontal shading; and(Sc5) shading with a linear tree arrangement. ENVI-met software version 5.1.1 and the RayMan model were used to estimate the local climate conditions and outdoor thermal comfort levels based on the physiological equivalent temperature (PET). All scenarios reduced PET values across the street, with optimal reductions of −2.0 °C, −3.1 °C, −1.3 °C, −1.7 °C, and −1.2 °C in Sc1, Sc2, Sc3, Sc4, and Sc5, respectively. Concerning pedestrian areas, the optimal results durations were at the southwest side below the arcades' sidewalks during peak hours: Sc2, Sc3, Sc4, Sc5 (2.2 °C–3 H, 2.3 °C–3 H, 2.4 °C–3 H, 2.5 °C–2 H). Sc1 performed best during daytime hours on the northeast side. The utilizing of these results can strongly help urban planners and landscape architects in creating climate-responsive streets that enhance citizens' quality of life.

**Keywords:** boulevard; heat stress; mitigation strategies; physiological equivalent temperature; spatial configuration; ENVI-met; walkability

## 1. Introduction

In the context of designing healthy cities, promoting walkability among interconnected spaces is considered a highly effective approach to achieving both sustainability and well-being. Sustainability encompasses efforts to mitigate the impact of air pollution and greenhouse gas emissions, as well as to address the effects of the urban heat island (UHI) [1–4]. Additionally, it concerns itself with the well-being, in term of promoting the physical health of residents and social interactions [5–8]. Therefore, many studies describe the urban environment parameters that help to stimulate human perception of walking, such as land use mix, street connectivity, human scale considerations, transparency, and spatial complexity [9–11]. Nevertheless, the provision of an urban environment that ensures a basic level of thermal

comfort is considered the critical factor in such a choice [4,12,13]. Local climate characteristics significantly influence outdoor thermal comfort (OTC), whereas urbanization alters these traits [14,15]. Urban elements affect radiative fluxes, the flow of wind and its intensity, thereby altering the thermal characteristics of urban spaces and pedestrian comfort within cities [16,17]. In this regard, urban planners can play a pivotal role in contributing to the establishment of a comfortable microclimate by adapting climate-responsive strategies [18–21].

### 1.1. Outdoor Thermal Comfort in Urban Canyon Studies

In recent years, there has been a large interest in studying the built environment and its relationship to OTC and the use of outdoor spaces [22]. In relevant studies, the built environment is often described under urban street configurations, considered as the fundamental structural element of the urban fabric [20,23,24]. These streets' collectivity accounts for a significant part of the city's area [25]. When considering climate-responsive urban design, the characteristics of these streets, also known as urban canyons, are adopted as a framework to describe the urban canopy within microscale conditions and as building blocks to describe the larger urban surface at the mesoscale [26].

In this context, urban canyons are defined by two fundamental properties: the height-to-width ratio (H/W) and the longitudinal axis orientation [26–28]. It has been demonstrated that these properties directly affect solar access and shading strategies, wind frequency, and heat exchanges that control surface and air temperatures [22,25,29,30]. As a result, they not only influence the OTC for individuals but can strongly impact the thermal comfort within indoor spaces, energy consumption, and cooling and heating demands [19,31–34].

The urban canyon is described as homogeneous if the average height of the building on two sides is approximately equal to the width (aspect ratio H/W $\approx$ 1). Thus, it is described as a shallow street if the aspect ratio (H/W) is less than 1, and a deep street if it is greater [25,35]. Therefore, studies conducted in hot climates focusing on the summer period confirm that, regardless of orientation, deep canyons with H/W $\geq$ 2 are preferred for improving OTC levels due to the amount of shade provided by the canyon [21,24,36–40]. In urban design, these results have led to the recommendation for using dense urban fabrics [18,41]. The functional organization of cities requires a hierarchical arrangement of streets [42]. Therefore, adopting wide streets, even in a limited proportion, is essential in designs. Furthermore, it is crucial to incorporate wider streets when discussing walkability in the city, to provide a functional mix of services that attract pedestrians [9]. The objective of this study lies in the examination of the main streets, often defined as shallow canyons in a climate-responsive urban design. The study focuses on the potential for walking, the use of streets, and pedestrian-specific areas.

In this regard, one of the main streets in the city of Biskra, Algeria, has been investigated to evaluate the thermal comfort levels of pedestrian areas within shallow canyons. The aim is to develop an optimal or nearly optimal design model for main streets. To achieve this objective, the current research is addressing the following questions: (a) How can design strategies be implemented to mitigate the impact of summer heat stress on pedestrian areas along main streets in hot desert climates? (b) What factors influence the appropriate placement of these strategies within the street layout?

### 1.2. Literature Review

Upon review of the literature on mitigating local climate effects in outdoor urban spaces during hot summer conditions, various possibilities of passive strategies have been identified. These approaches encompass interventions in urban geometry, shading techniques, urban vegetation, the use of high-albedo materials, and the incorporation of water bodies [22,25,30,41,43]. Regarding urban canyons, several scenarios for interventions in urban geometry involve adopting symmetrical street profiles, while modifying the aspect ratio and orientation [23,24]. In this regard, previous reports have emphasized the necessity for implementing deep canyons with a north-south (N-S) orientation or other

intermediate orientations (northwest-southeast (NW-SE) and northeast-southwest (NE-SW)) [22]. However, very limited studies have assessed the impact of asymmetrical profiles on OTC [20,21,37,44].

In the case of streets with asymmetrical shapes, it has been reported that the thermal conditions in pedestrian areas differ compared to streets with symmetrical shapes [21,44]. Rodríguez-Algeciras et al. (2018) conducted a study on pedestrian areas in asymmetric streets and observed that pedestrian conditions vary on the side of the street related to the position of the highest facades and the street orientation [21]. This study was carried out on a medium-width street (9 m) and focused on Cuba's hot and humid climate. Therefore, it should be noted that buildings' projections along the street borders and their direct impact on pedestrian space were not extensively studied. The study by Ali-Toudert and Mayer (2007) discussed the impact of galleries and emphasized their positive effect on reducing the duration of thermal stress [37].

Various shading strategies are employed in public spaces, including horizontal, vertical, and inclined elements, membrane roofs, and photovoltaic structures [39]. However, research on urban canyons primarily revolves around evaluating buildings and tree shading [39,45], with limited studies delving into supplementary strategies. In this regard, Swaid and Hoffman (1990) affirmed that, in the Mediterranean climate, by introducing a vertical screen on the south-facing wall within an E-W street, solar irradiance can be reduced by approximately 150 W/m$^2$, and air temperature (Ta) can decrease at noon in the summer season [46]. Furthermore, installing textile shading canopies contributed to a surface temperature reduction of 16 °C on sidewalks inside an E-W street in Cordoba, Spain [47]. Nonetheless, none of these studies have directly investigated the impact of these strategies on the OTC. However, research focused on street trees [41,48–50]. It has been emphasized that street trees can reduce mean radiant temperature (Tmrt) and (Ta), especially in wide canyons, and enhance the OTC in two fundamental aspects: shading and evapotranspiration [25]. Trees could have an inverse effect in deep canyons [51] and during nighttime hours [52]. However, the effect of trees depends on their type, size [53], position regarding buildings and adjacent trees [54], and their canopy characteristics [55,56].

On the other hand, numerous research studies have been conducted on high-albedo materials as a potential remedy for enhancing outdoor thermal conditions. These materials possess thermophysical characteristics that enable them to absorb minimal solar radiation, leading to reduced heat retention and lower surface temperatures. Consequently, the emission of long-wave radiation toward the surroundings is diminished, contributing to a decrease in (Ta) levels [30]. Employing cool materials exclusively on building rooftops can indeed decrease the Ta thresholds above these surfaces and mitigate the UHI effect by up to 0.3 K [57]. However, the impact on cooling pedestrian areas remains relatively limited, depending on the height of the buildings [58,59].

The implementation of cool materials on horizontal ground surfaces in wide urban canyons led to a reduction of Ta by 0.75 K, 3.5 K, and 6.4 K, respectively [60–62]. Otherwise, their use in deep canyons (H/W > 1) did not yield any observable effect due to the significant shading within the canyon [30,32]. High-albedo materials on vertical building facades and horizontal ground surfaces can lead to the opposite effect. This may result in a rise in Ta due to the significant increase in short-wave reflections within street canyons [60].

Generally, the implementation of high-albedo materials can decrease Ta related to the reduction in the emission of long-wave radiation. However, this same implementation can elevate Tmrt levels due to the increase in the emission of short-wave radiation. This latter can directly affect the radiative exchange between pedestrians and the built environment, increasing thermal stress [22,30]. In this regard, some research showed opposing results; Rosso et al. (2018) assessed the urban canyons in a historic district (aspect ratio of 3.5) and confirmed that using high-albedo pavements alongside low-albedo walls reduced the thermal stress of pedestrians [63]. Therefore, researchers emphasize that the spatial arrangement of the study area plays a crucial role in determining the impact of this strategy [24,64].

Accordingly, coupling highly reflective materials and street trees in specific layouts can achieve two goals: reducing Ta and Tmrt, and improving OTC [30,32].

Due to the literature's focus on street activity and planning forms, water bodies and grass surfaces have not been adequately addressed as potential mitigation strategies inside these urban spaces.

The literature investigates the mitigation strategies within various geometric configurations of streets and adopts case studies to compare and evaluate the implemented planning methods. The originality of this research lies in coupling several strategies to assess design alternatives for a specific street pattern. Furthermore, the authors consider various areas of the street to compare the performance and best position of potential solutions, to achieve an optimal or nearly optimal design model. Through a more comprehensive perspective, this work can establish a methodological guideline for addressing street design aspects to make them more walkable under suitable thermal conditions.

The current study was conducted on 'Emir Abdelkader Boulevard' in Biskra city and consisted of two essential phases. Firstly, it involved evaluating the current thermal comfort level during the summer. Secondly, it entailed assessing modeled mitigation scenarios derived from the previous relevant studies.

## 2. Methodology

The current study combines two interrelated approaches to achieve its objectives: field measurements and simulation. Figure 1 provides a comprehensive representation of the study's conceptual framework, outlining the primary steps of the research methodology. Overall, the field measurements enabled accurate validation of the ENVI-met software's models for use with climatic data within the study context. The validation step allowed the software to be employed to assess the modeled scenarios involving asymmetrical profiles, cool pavement albedo, and shading, which can influence pedestrian thermal comfort conditions. The PET index was calculated using the RayMan model based on ENVI-met output parameters used for this purpose.

### 2.1. Study Context and Selected Area Description

The investigation was carried out under typical summer day conditions in Biskra city, located at latitude 34°48′ north and 5°44′ east, Southern Algeria. According to the Köppen–Geiger climate classification, Biskra belongs to the hot desert climate zone (BWh) [65], characterized by a hot and dry climate, with an average annual temperature of 21.8 °C and an annual precipitation of less than 141 mm [66]. Moreover, July and August present the hottest months of the year, with a seasonal average of 40 °C [67]. The Algerian desert has experienced an increase in heatwave days over the past three decades [68]. This phenomenon poses health risks to individuals in outdoor environments. Meteorological data from Biskra Airport indicate that cooling is necessary for six months, spanning from May to October. On the other hand, from November to April, Biskra experiences a cool season as its winter, characterized by mild and lower temperatures, along with occasional rainfall [66].

The chosen site of Boulevard 'Emir Abdelkader' is considered an extension of National Road No. 31, which crosses the city center. Together with 'Zaatacha Boulevard', they form the two main axes of the Biskra city center (Figure 2b). During favorable weather conditions, the boulevard experiences an important flow of pedestrians due to the various services and markets available on both sides, including shops, local administration, hotels, and restaurants. With its historical position, 'Emir Abdelkader Boulevard' is the oldest boulevard in the city, featuring a mix of French and modern buildings. Furthermore, its length reaches around 750 m, and along 400 m, the street is lined by arcades (covered sidewalks) that are 4 m in width on both sides, interspersed with areas featuring arcades on one side. Therefore, the width between the two boulevards' facades reaches 16 m and 20 m, respectively (see Figure 2c). The average building height is 10 m, with variations

ranging from two-story to seven-story buildings. Therefore, the aspect ratio (H/W) varies between 0.5 and 0.62.

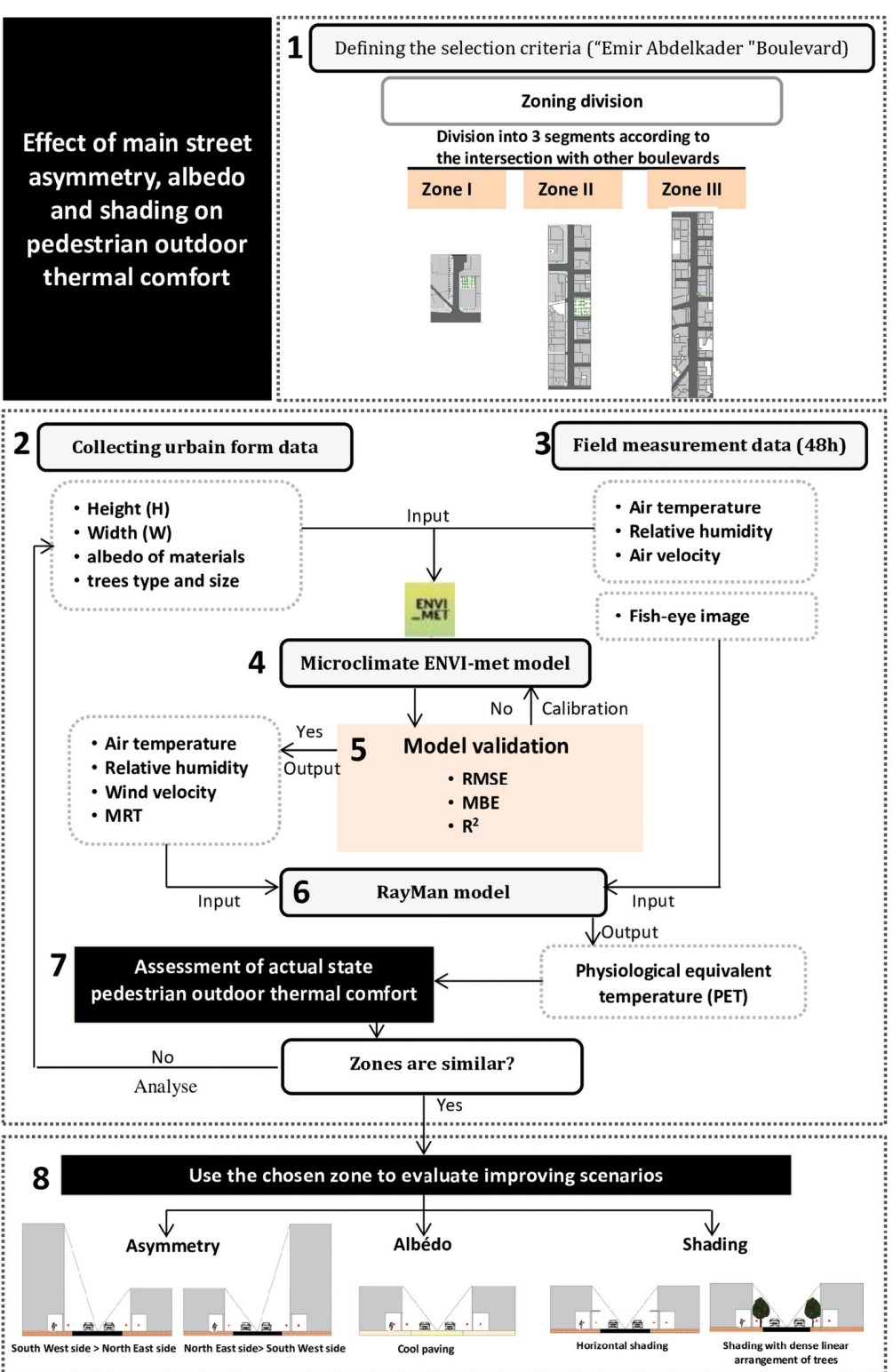

**Figure 1.** Study conceptual framework.

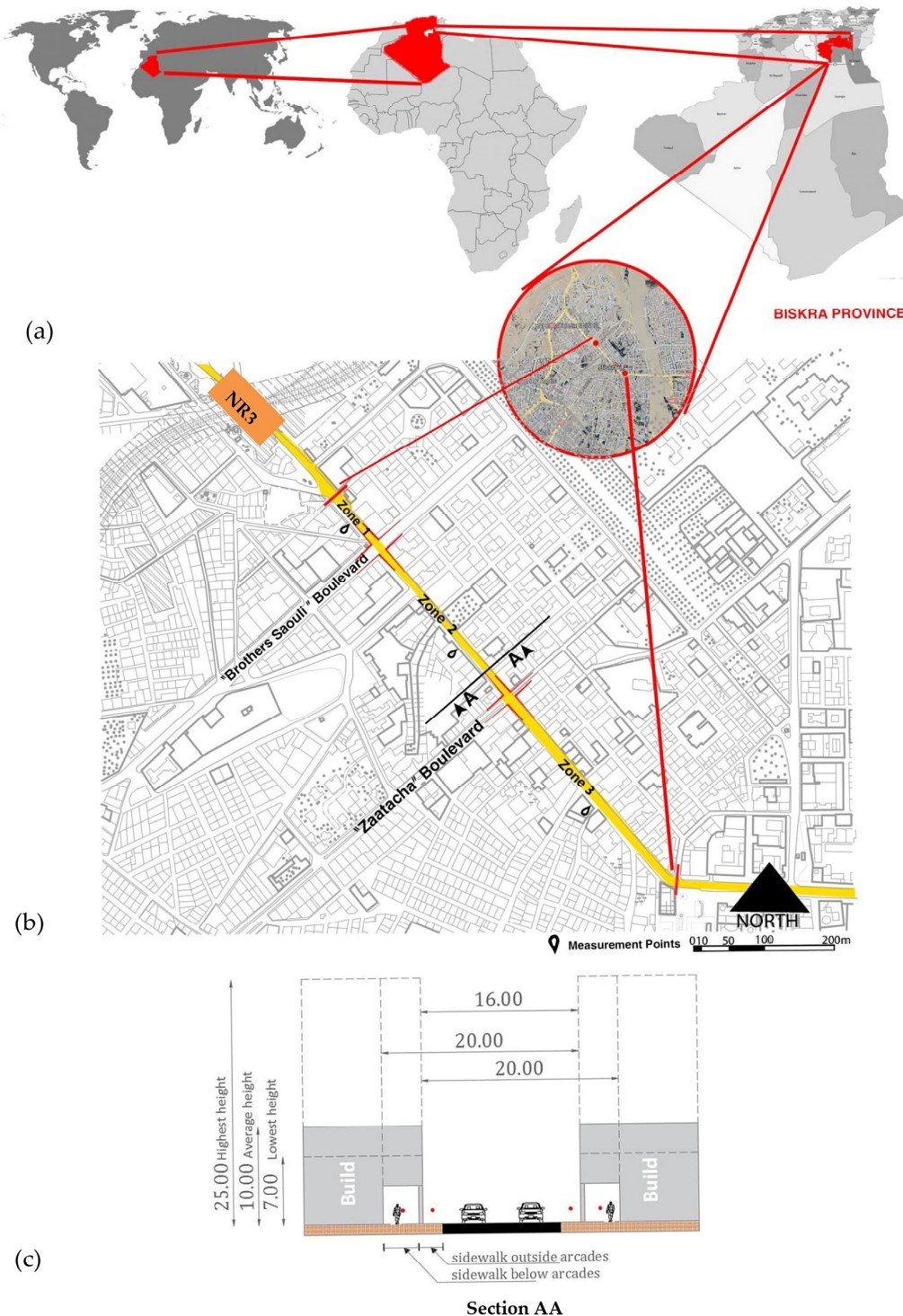

**Figure 2.** Study area identification: (**a**) Location of Biskra province. (**b**) Location of 'Emir Abdelkader Boulevard', defining of zoning and measurement points. (**c**) Representative section of the Boulevard.

### 2.1.1. Defining the Selection Criteria

The boulevard 'Emir Abdelkader' was chosen for several criteria:

- The width represents the mean width used throughout the main streets in Algerian desert cities. Previous studies have not recorded an aspect ratio of less than 0.5 in hot and arid climates [24,38,39,69,70].
- Northwest—southeast orientations are highly recommended for these climate areas [70].

- The presence of arcades (covered sidewalks) on both sides is strongly recommended within Algerian urban planning regulations specifically for main streets.

### 2.1.2. Zoning Division

For better control over simulation stages and a clear understanding of the results, the street has been divided into three zones based on the intersections with other boulevards (see Figure 2b):

Zone 1: From the beginning of the street to the intersection with 'Brothers Saouli Boulevard'.

Zone 2: From the intersection with 'Brothers Saouli Boulevard' to the intersection with 'Zaatacha Boulevard'.

Zone 3: From the intersection with 'Zaatacha Boulevard' to the end of the street (east–west diversion).

### 2.2. Field Measurement and Collecting Data

In many studies, the primary objective of field measurements is to validate the created numerical models [22,30]. For this purpose, three points were selected on pedestrian sidewalks outside the arcades (Figure 2b). The measurements were conducted over 48 h on two typical summer days (7 and 8 August 2021). Moreover, the Testo 480 datalogger was utilized for measurements, which is a multifunctional device equipped with digital probes, each individually calibrated. This instrument was used sequentially as a thermo-hygrometer and then as an anemometer. To measure air temperature (Ta) and relative humidity (R.H), a specific probe (Ø 12 mm) with a high resolution of 0.01 and an accuracy of ±0.03% R.H and ±0.2 °C was utilized. For air velocity (Va), a fan-assisted measuring probe (Ø 16 mm) was used, providing an accuracy of ±0.2 m/s. The measurements were taken at a height of 1.4 m. The obtained hourly data include Ta (°C), R.H (%) and Va (m/s).

The data on building heights, construction materials, and types and canopies of trees were obtained through a field survey in the study area.

### 2.3. Microclimate ENVI-Met Software

This study requires integrative spatial assessment software to evaluate the impact of physical parameters within the urban street on the OTC. ENVI-met software has high sensitivity in estimating outdoor microclimate conditions [29,71–74]. It employs a set of algorithms and mathematical models based on the fundamental laws of Computational Fluid Dynamics (CFD) to generate precise 3D simulations of microclimates [30]. By forecasting dynamic interactions between the atmosphere, Earth's surface, and environmental objects (buildings, pavement surfaces, plants and sources of pollution) [75], the software offers an in-depth analysis of air temperatures, humidity, wind speed, mean radiant temperature, solar radiation, and other crucial climatic parameters [76,77]. The simulations provide detailed insights into the spatial and temporal parameters' distribution at a high quality of resolution varying from 0.5 to 10 m and a time step rate of 10 s [30,78,79].

ENVI-met software is widely used in the fields of microclimate and thermal comfort analysis [23,80–84]. The reliability of the software in assessing hot climates has been approved in numerous research studies [32,37,39,85–87]. Nevertheless, it remains crucial to pay attention to limitations identified in previous studies, specifically the issue of underestimating diurnal temperature variations [59,88–91]. Otherwise, it should be noted that the physical processes within the model tend to stabilize after the first 24 h of running the simulation, leading to improved accuracy in simulating the following 24 h [23,82]. Therefore, the simulation duration in this study was set to 48 h, while the results section summarized only the second 24 h of the simulated models.

### 2.3.1. ENVI-Met Settings and Details of Input Data

This study used the latest scientific version of ENVI-met 5.1.1. To verify the model's accuracy, the measurement data recorded from the three investigative zones were used as

input data in the configuration. The arrangement of the zones was designed according to a drawing supplied by the Municipality of Biskra. Building heights and tree species were derived from field observations. Despite their scarcity, all existing street-lining trees were simulated to assess the current situation. The Albero 5.1.1 model's options were employed to model these trees, and the selection of modeled trees can be directly made from the model's SPACE. To ensure the stability of the simulation's grid boundary conditions, a "nesting grid" with a minimum height of twice the highest building was used in the horizontal boundaries [92]. Additionally, the distance between the top boundary of the model area and the ground level was set to three times the height of the highest building [23]. Accordingly, the configurations for all numerical model inputs are shown in Table 1.

**Table 1.** ENVI-met settings and details of model input data.

| | Zone 1 | Zone 2 | Zone 3 | Scenarios | | | | | |
| | | | | SC0 | SC1 | SC2 | SC3 | SC4 | SC5 |
|---|---|---|---|---|---|---|---|---|---|
| Model location | | | | Biskra (34°48′ N; 5°44′ E) | | | | | |
| Domain size (m): | | | | | | | | | |
| X Direction | 100 | 100 | 100 | 100 | 100 | 100 | 100 | 100 | 100 |
| y Direction | 116 | 304 | 360 | 304 | 304 | 304 | 304 | 304 | 304 |
| Z Direction | 30 | 40 | 50 | 30 | 50 | 50 | 30 | 30 | 30 |
| Spatial resolution (X, Y, Z) | | | | 2 m × 2 m × 2 m (in Z direction, lowest grid box is split into 5 sub-cells) | | | | | |
| Model rotation | | | | 315° | | | | | |
| Total simulation time | | | | 48 h | | | | | |
| Date of simulation | | | | 7 and 8 August 2021 | | | | | |
| Meteorological inputs | | | | Full forcing (CSV file) | | | | | |
| Building material | | | | Wall: hollow block concrete; Roof: cast dense concrete | | | | | |
| Road soil | Asphalt. | Asphalt. | Asphalt. | Asphalt. | Asphalt. | Asphalt. | Stamped light concrete | Asphalt. | Asphalt. |
| Sidewalk's soil | Brick yellow stone | Brick yellow stone | Brick yellow stone | Brick yellow stone | Brick yellow stone | Brick yellow stone | Concrete pavement light | Brick yellow stone | Brick yellow stone |
| Alignment trees | New deciduous Trees: Spherical (15 m); Palm Trees: medium (15 m); Ficus Retusa Trees: medium (15 m) | | | / | / | / | / | / | Ficus Retusa Trees: medium (15 m) |

On the other hand, the building materials used in the 'Emir Abdelkader Boulevard' model are concrete for buildings, Asphalt Road (S.T) for car roads, Brick yellow stone (K.G) for pedestrian pavements, Loamy Soil (L.O) for "nesting grid ", and natural surfaces.

### 2.3.2. ENVI-Met Output Data

In the assessment phase, the same field measurement points were selected for data processing (see Figure 3). However, in the scenarios' evaluation phase, the following points were taken:

- First: obtain the average values to assess the impact of the designed scenarios on the entire study area.
- Second: obtain receptor values defined to assess the impact of scenarios on pedestrian areas within four principal points—(A) the southwest sidewalk below the arcades; (B); southwest sidewalk outside the arcades; (C) northeast sidewalk outside the arcades; (D) northeast sidewalk below the arcades; and (E) center of the street (for comparison).
- Third: determine the values for each grid unit within a representative section of the street to draw the spatiotemporal distribution of the scenarios' impact.

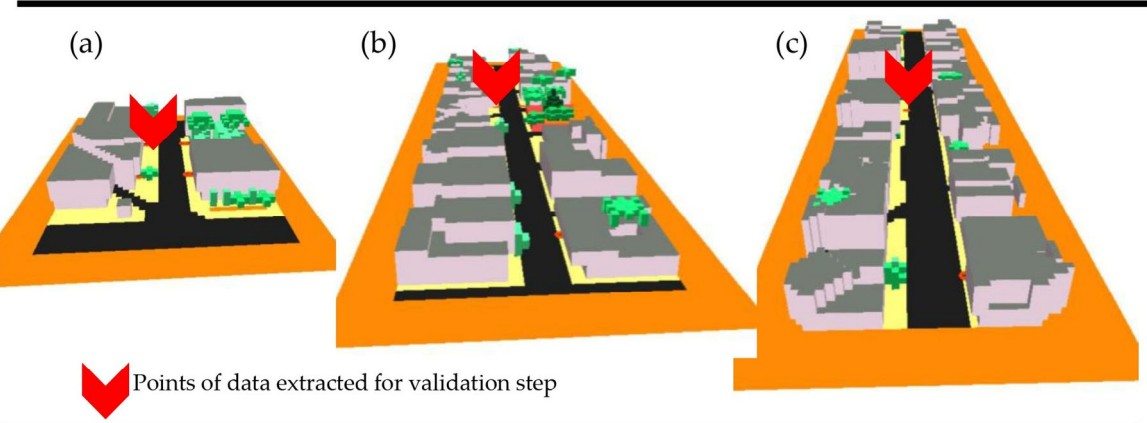

**Figure 3.** Points of extraction data for evaluation and validation steps: (**a**) zone 1, (**b**) zone 2, (**c**) zone 3.

*2.4. ENVI-Met Validation*

The validation of the numerical model takes into account both the simulated data and the measured datasets taken on-site [93]. Therefore, during hot summer conditions, many studies rely on the (Ta) as a critical parameter for the model's validation [30]. According to ASHRAE 14-2014 guidelines [94], it is recommended to combine the root mean square error (RMSE) with the mean bias error (MBE) as statistical metrics to evaluate the accuracy of ENVI-met numerical models. Furthermore, the previous literature emphasized the importance of the coefficient of determination ($R^2$) in visualizing the dispersion of compared values concerning the reference regression line on the graph [30,95,96].

$$\text{RMSE} = \sqrt{\frac{1}{n} \cdot \sum_{i=1}^{n} \left(\text{Sim}_i - \text{Obs}_i\right)^2} \ (\%) \tag{1}$$

$$\text{MBE} = \frac{1}{n} \cdot \sum_{i=1}^{n} \left(\text{Sim}_i - \text{Obs}_i\right) \ (\%) \tag{2}$$

Table 2 illustrates the validation results of the numerical models, showing their high reliability for all three study zones. In general, it should be noted that the model tends to underestimate the (Ta) throughout the simulation period (Figure 4), with average differences between measurement and simulation of less than 1 °C (0.82 °C in zone 1, 0.76 °C in zone 2, and 0.68 °C in zone 3). According to previous studies, these values are considered very acceptable [39,97–102]. Neglecting non-permanent heat sources, such as external heat from air conditioners and vehicles, may contribute to this underestimation. However, the validation results confirm the ENVI-met model's accuracy and provide confidence in the results for further analysis.

**Table 2.** Validation of the simulated models: statistical metrics values.

| Indices | Emir Abdelkader Boulevard | | | | | |
|---|---|---|---|---|---|---|
| | Zone 1 | | Zone 2 | | Zone 3 | |
| RMSE | 0.91 | 2.26% | 0.84 | 2.10% | 0.74 | 1.38% |
| MBE | −0.83 | −2.07% | −0.78 | −1.95% | −0.69 | −1.71% |
| R² | 0.9928 | | 0.9923 | | 0.9953 | |

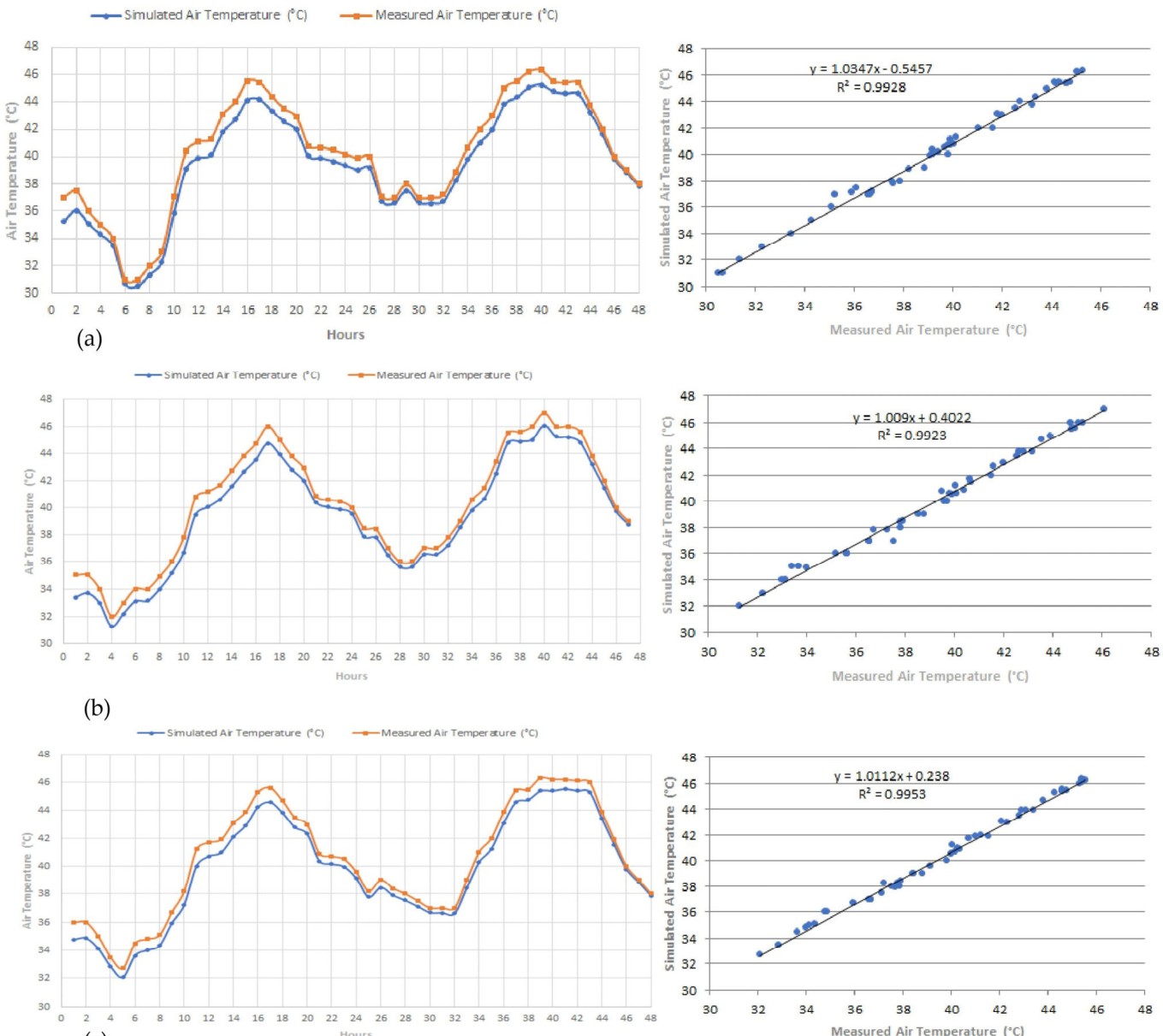

**Figure 4.** Temporal patterns of measured and simulated (Ta) and coefficient of determination ($R^2$) graph from (**a**) zone 1, (**b**) zone 2, (**c**) zone 3.

## 2.5. PET and RayMan Model to Assess the OTC

The assessment of the baseline model and the mitigated scenarios was based on the physiologically equivalent temperature (PET), which is a widely adopted index to evaluate thermal comfort at the urban level [103]. In the current study, the calculation of the PET index was performed by the RayMan Pro 3.1 Beta model using the ENVI-met simulation

outputs. RayMan is a microscale software specifically designed for environmental meteorology [104,105]. This model aims to determine the levels of short- and long-wave radiation, sunshine duration, and relevant assessment indices related to thermal comfort in intricate urban settings [106]. A minimal set of microclimate data is required, including (Ta, R.H, Va, Tmrt), geographic coordinates of the study area, and the geometric characteristics of the environment, such as the sky view factor (SVF) at the reference location.

PET is a thermal comfort index that relies on a diagnostic model of the human body's energy balance. Therefore, it considers personal factors such as individuals' activity levels, clothing insulation, and age [107]. In the study case, the settings were established as follows: a 35-year-old male weighing 75 kg and measuring 1.75 m in height. The clothing insulation was set at 0.9 clo, and the metabolic rate was assumed to be 80 W/m$^2$ due to light walking activity. The decision to use constant values for clothing and activity in calculating PET was made to create an index that does not rely on individual behavior [106,108].

*2.6. Baseline Model Assessment*

To analyze the thermal comfort levels of pedestrians on 'Emir Abdelkader Boulevard' during two typical summer days, the PET index was calculated using simulation output parameters via ENVI-met. This calculation was combined with fisheye images taken on-site at a height of 1.4 m. The resulting data were used as inputs in the RayMan model. Furthermore, the PET results were based on the modified PET scale adapted to an arid climate (BWh), as outlined in the study by Cohen et al. (2019) (Table 3) [109]. The results shown in Figure 5 indicate a significant similarity among the three zones, with thermal levels ranging from slightly warm to very hot, passing through warm and hot levels. Surprisingly, there are no comfortable periods, even during the night. Consequently, peak temperatures exceeded 42 °C, resulting in extreme heat stress, which started at 09:00 until 19:00, and it happened similarly on the second day (8 August 2021).

**Table 3.** Adjusted (PET) scale for (BWh) climate [109].

| PET (°C) | Thermal Sensitivity | Grade of Thermal Stress |
| --- | --- | --- |
| >42.0 | Very hot | Extreme heat stress |
| 37.1–42.0 | Hot | Strong heat stress |
| 28.1–37.0 | Warm | Moderate heat stress |
| 26.1–28.0 | Slightly warm | Slight heat stress |
| 17.1–26.0 | Neutral | No thermal stress |
| 13.1–17.0 | Slightly cool | Slight cold stress |
| 8.1–13.0 | Cool | Moderate cold stress |
| 6.1–8 | Cold | Strong cold stress |
| <6.0 | Very cold | Extreme cold stress |

Consequently, zone 2 was identified as the most suitable area for implementing mitigation scenarios. Zone 2 was chosen due to its having the lowest average PET values during the selected two days.

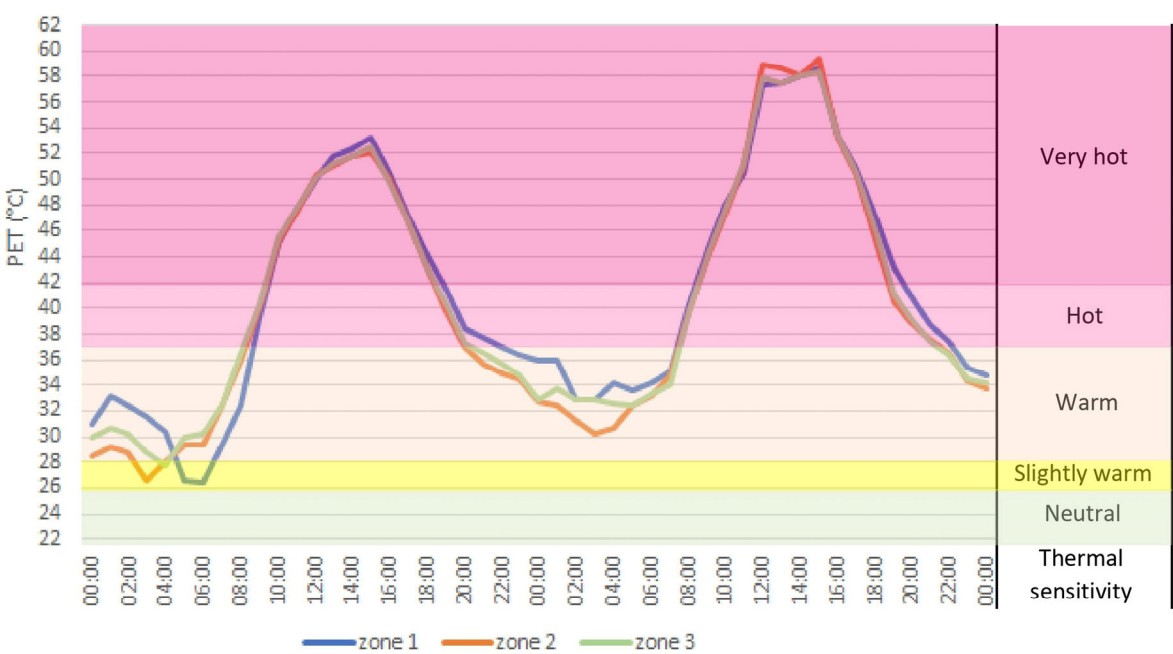

**Figure 5.** PET level in 'Emir Abdelkader Boulevard' during study period (7 and 8 August 2021).

### 2.7. Mitigation Scenarios

The actual spatial arrangement of zone 2 was maintained as the baseline scenario. In order to standardize the design principle along the street and make the results more comparable, a reference scenario (SC0) was designed. The (SC0) scenario contains buildings, arcades, and sidewalks that have been aligned, and the average building height of 10 m is maintained as a unified building height (see Figure 6).

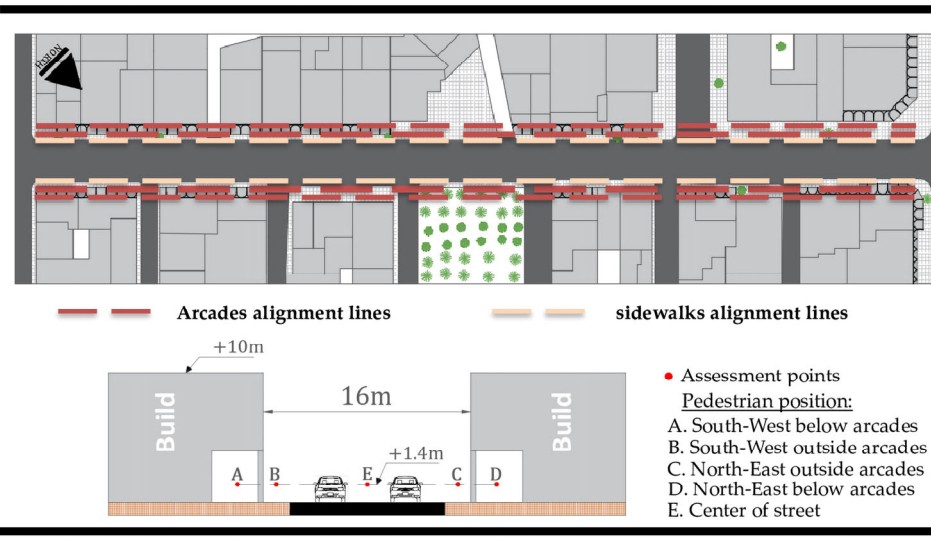

**Figure 6.** Modified parameters for the reference scenario (SC0).

Therefore, five scenarios were proposed depending on the main factors of the study. Detailed characteristics and the respective features of these scenarios can be found in Table 4.

**Table 4.** Proposed mitigation scenarios.

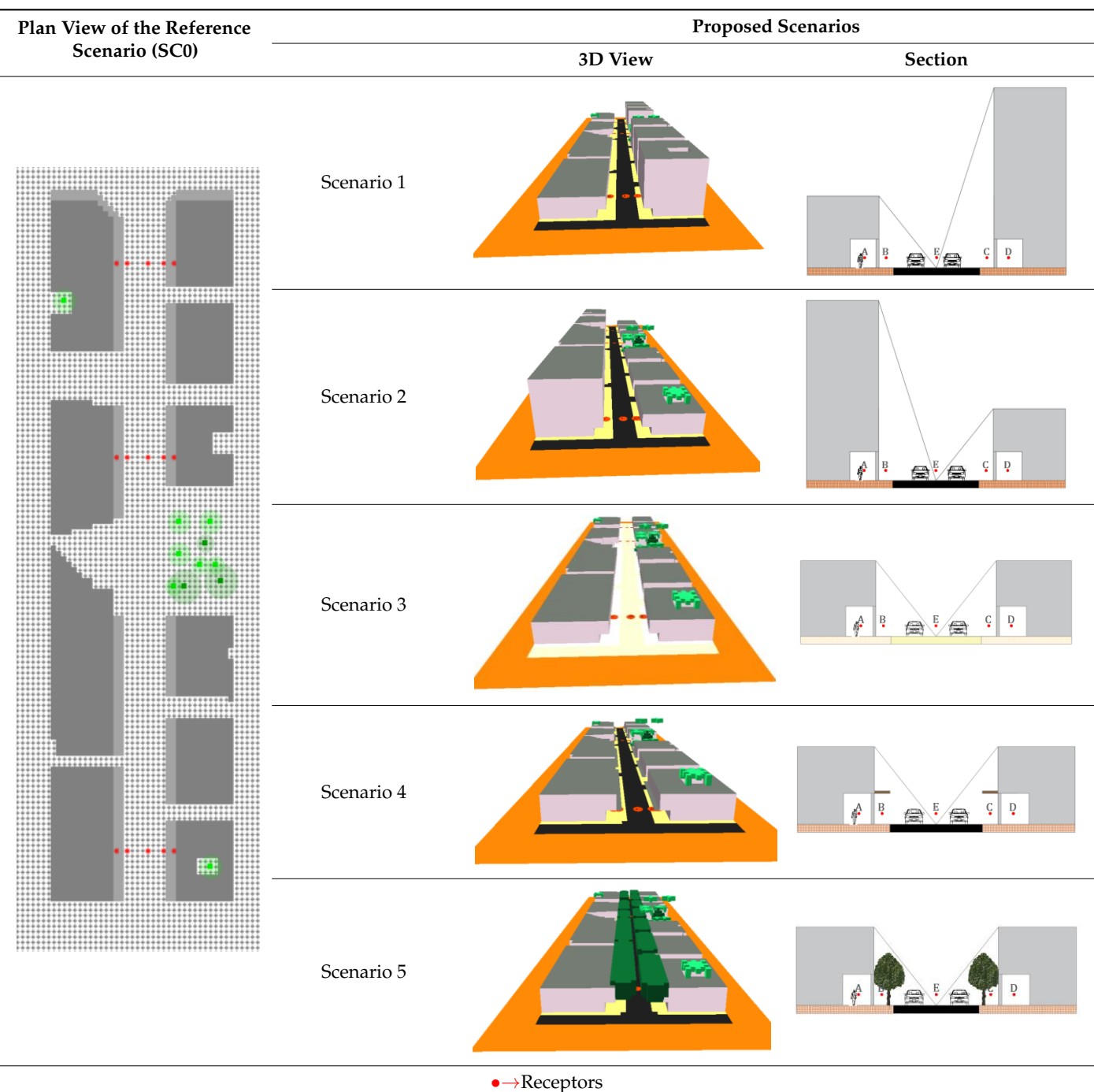

1. Scenario 1 (SC1): This features an asymmetric profile, where the northeast side represents the tallest buildings' side of the boulevard and the southwest takes the mean height.
2. Scenario 2 (SC2): Inversion of the asymmetric sides of the scenario 1 (SC1).
3. Scenario 3 (SC3): Utilization of cool pavement. Thus, the road is stamped by light concrete pavement with an albedo of 0.5 and emissivity of 0.9, whereas the sidewalks are covered by light concrete pavement with an albedo of 0.8 and emissivity of 0.9.
4. Scenario 4 (SC4): A horizontal canopy has been designed to cover the sidewalks outside the arcades, covering the surface beyond the arcades.

5. Scenario 5 (SC5): Implementation of Ficus Retusa, which is a commonly used tree species in urban arrangements in the region [50]. These trees, about 12 m in height (modeled by Albero), have been aligned on both sides of the boulevard within the sidewalks outside the arcades.

## 3. Results and Discussion

Due to significant variations in Ta, Tmrt, and PET index, the study places considerable focus on these parameters. Generally, air velocity (Va) and relative humidity (R.H) values are lower during August (Va < 3 m/s and R.H < 25%). As mentioned before, all the obtained results were taken at pedestrian level, at a height of 1.4 m.

### 3.1. Scenarios Effect in Thermal Conditions of the Study Area

The simulation results of (Ta) throughout the study area indicate a high pedestrian heat stress. Furthermore, in the reference scenario (SC0), the Ta rises above the 40 °C threshold from 10:00 to 20:00, reaching its peak at 15:00, when Ta exceeds 46 °C.

On the other hand, the proposed scenarios show a remarkable decrease in temperatures during the daytime hours compared to nighttime hours (see Figure 7). SC5 shows the most favorable results, with an average reduction in Ta equal to 0.9 °C during the daytime hours and 0.3 °C at the nighttime hours. The highest reduction value occurred at noon with a value of 1.36 °C. This obtained result can be attributed to the beneficial impact of trees in reducing the air temperature via dense shading within wide streets. Moreover, the second-best result was obtained in SC2, with an average Ta equal to 0.5 °C during the daytime hours. Otherwise, during the nighttime hours, the reduction's level was approximately 0.15 °C, which aligns with scenarios 3 (SC3) and 4 (SC4) during nighttime hours. SC3 and SC4 showed a similar daytime mitigation, with a slight preference for Scenario 4, with a decrease equal to 0.32 °C during daytime hours. However, SC1 shows the lowest Ta mitigation value during the daytime hours, with an average of 0.16 °C. One possible reason for this aspect is the increase in facade surface temperatures, exposed to solar radiation during the afternoon in SC1.

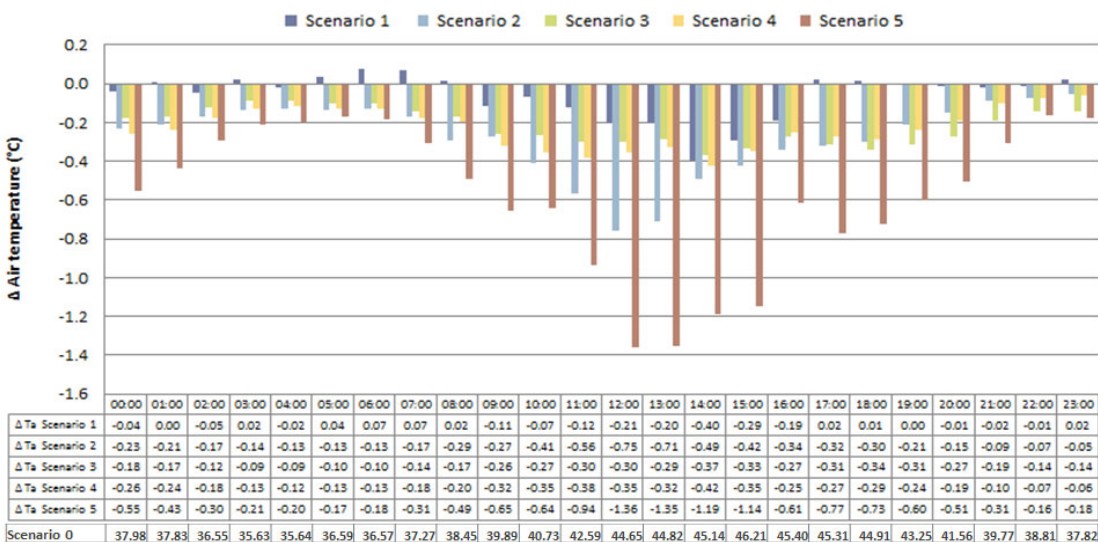

**Figure 7.** Δ average of air temperature between mitigation scenarios and reference scenarios.

Tmrt is one of the most important parameters regarding urban spaces. During the summer season, it becomes the key factor in determining the level of human outdoor comfort (OTC). Figure 8 illustrates the hourly variations in Tmrt averages in the reference scenario (SC0). Tmrt remains above 50 °C during 8 h (from 10:00 to 17:00), reaching approximately 70 °C at 15:00.

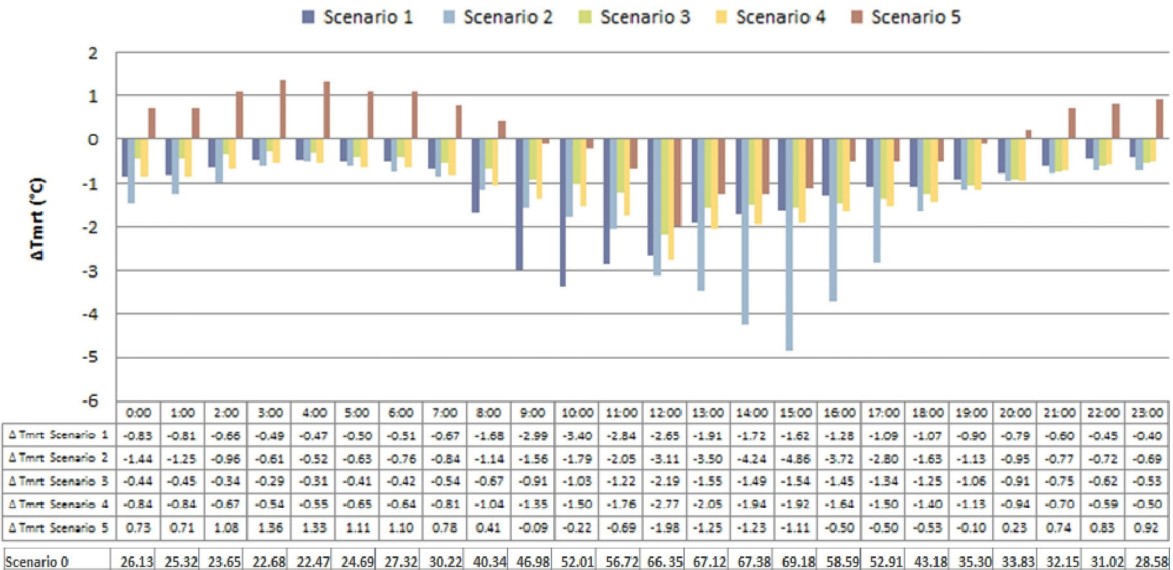

**Figure 8.** Δ average Tmrt of study area between mitigation scenarios and reference scenarios.

On the other hand, during the daytime hours, the scenarios exhibit performance in reducing Tmrt values that are directly related to the shaded surfaces compared to SC0. SC1 and SC2 show the optimal values for reducing Tmrt, with variation between morning and afternoon. Compared to SC0, the maximum reducing of Tmrt that occurred in (SC1) was equal to −3.40 °C at 10:00, whereas the maximum reduction was achieved by (SC2) with −4.86 °C at 15:00. Otherwise, the highest reduction values for SC3, SC4, and SC5 were obtained at noon hours with 2.19 °C, 2.77 °C, and 1.98 °C, respectively. The reduction values decrease gradually during the morning and afternoon periods, depending on the global solar radiation. Therefore, with shading scenarios, the SC4 recorded the best results compared to SC5. Consequently, this result is attributed to the fact of tree properties, such as the canopy, that directly influence the reflection of direct solar radiation. However, a flat surface can reflect more solar radiation flux than a tree canopy, leading to an effective solar radiation decrease in SC4. Otherwise, during the nighttime hours, SC5 showed the worst results, with an average increase in Tmrt of approximately 1 °C. This is probably due to the tree canopies' ability to reduce outgoing long-wave radiation and retain diurnal heat, generating an increase in nighttime Tmrt values.

The calculation of the PET index encompasses all the previous microclimatic parameters regarding human thermal perception. Therefore, in SC0, PET values remain at a level of extreme heat stress (PET > 42 °C) from 09:00 to 18:00, reaching a peak of 59.6 °C at 15:00.

Overall, Figure 9 indicates that all the proposed scenarios notably enhanced the PET values throughout the entire study area. The effectiveness of scenarios SC1, SC2, SC3, and SC4 in reducing PET index values was directly relative to their performance in reducing Tmrt values. In this regard, SC1 and SC2 yielded the optimal results between the morning and afternoon periods (maximum reduction of PET values is equal to −2.0 °C at 10:00 in SC1 and −3.1 °C at 15:00 in SC2). Moreover, SC3 and SC4 achieved their optimal results at 12:00 (maximum reduction of PET values is equal to −1.3 in SC3 and −1.7 °C in SC4). Overall, the effectiveness of these four scenarios in improving thermal comfort during the day is strongly related to their ability to control direct solar radiation exposure. Otherwise, SC5, contrary to its performance regarding Tmrt, displayed an improvement in PET values during the afternoon period from 12:00 to 20:00, with an average reduction of 1.2 °C. This can be attributed to the trees' ability to lower the air temperature (Ta), as mentioned previously.

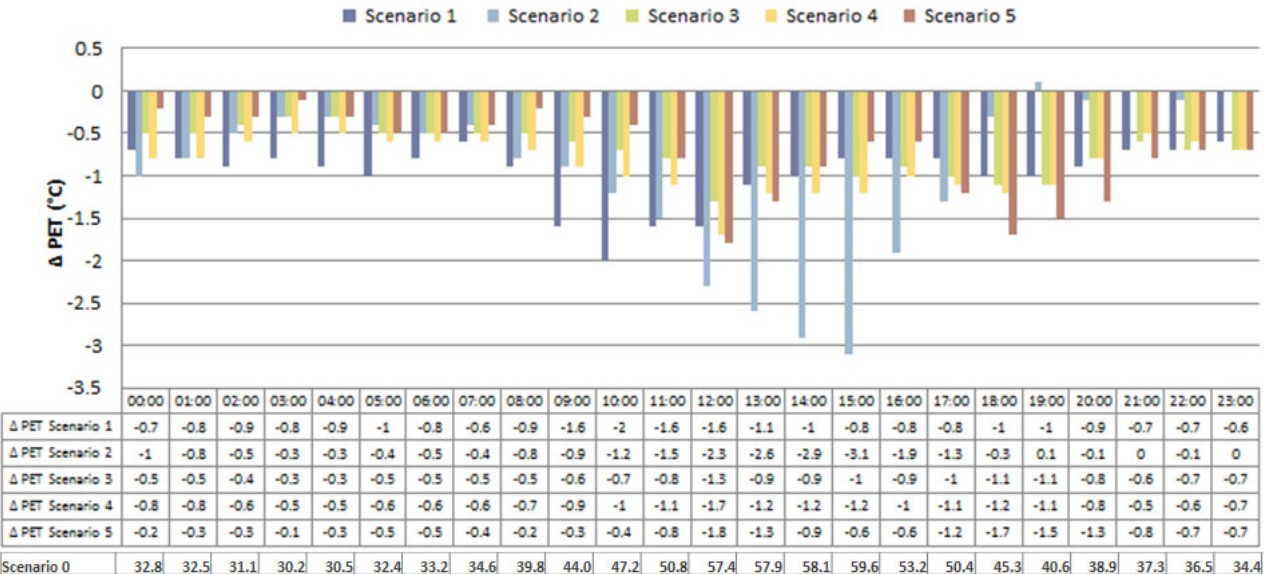

**Figure 9.** Δ Average PET of study area between mitigation scenarios and reference scenarios.

### 3.2. Scenarios Effect in OTC of Pedestrian Areas

The hourly PET index values were calculated during the second day of the study (8 August 2021) through specific points A, B, C, and D, which represent the principal pedestrian itinerary throughout the 'Emir Abdelkader Boulevard'. All the points are compared to point E, which is located in the middle of the boulevard. The results for each scenario were presented as box plots (see Figure 10). In this section, the analysis will be limited to the critical hours of heat stress. These critical hours correspond to PET values between the third quartile and maximum values of the box plots. Thus, the values represent 25% of the 24 h data, referring to the 8 critical hours of human heat stress in the study area.

In the reference scenario (SC0) (Figure 10a), point A representing the SW sidewalk below the arcades exhibits the most favorable conditions during the critical hours of the day, with a PET reduction varying from 2.8 °C to 5.3 °C versus point E, with 4.1 °C during 8 h. Moreover, point D in the NE sidewalk below the arcades showed an average decrease of 2.6 °C. Point B in the SW sidewalk outside the arcades showed an average decrease of 1.8 °C. Finally, point C in the NE sidewalk outside the arcades showed an average decrease of 1.5 °C.

In SC1 (Figure 10b), no significant impact was observed during the 8 critical hours in pedestrian zones compared to the baseline scenario (SC0). This result can be attributed to the effectiveness of this scenario, which is to be remarked upon during the early daytime hours, when the 8 critical hours primarily occur in the afternoon period. Otherwise, a slight improvement is obtained in point D, with a maximum of 0.6 °C, located below the highest building.

Scenario 2 (SC2) significantly affects the thermal conditions during the 8 critical hours. Compared to the baseline scenario (SC0), PET values are improved by up to 2.2 °C in point A, 1.5 °C in point B, 1.1 °C in points E and C, and 1 °C in point D. Therefore, in this scenario, the difference between point E and point A can reach 6.4 °C. This result can be attributed to the large shaded surfaces provided by the tallest buildings' sides during critical hours.

The impact of scenario 3 (SC3) through the covered points not receiving direct solar radiation was significant. Thus, at point A, PET values were improved by 2.3 °C compared to the reference scenario, while point D improved by 1.7 °C and point B by 1.3 °C. On the other hand, the mitigation was very limited in points C and E (0.3 °C). This result can be attributed to the high albedo's ability to reduce surface temperatures, thereby enhancing air temperature Ta. However, it also increases the reflection of short-wave radiation, which increases Tmrt values. Accordingly, in the points receiving direct solar radiation, the

reflection of short-wave radiation increases highly compared to other points, resulting in less perceived mitigation.

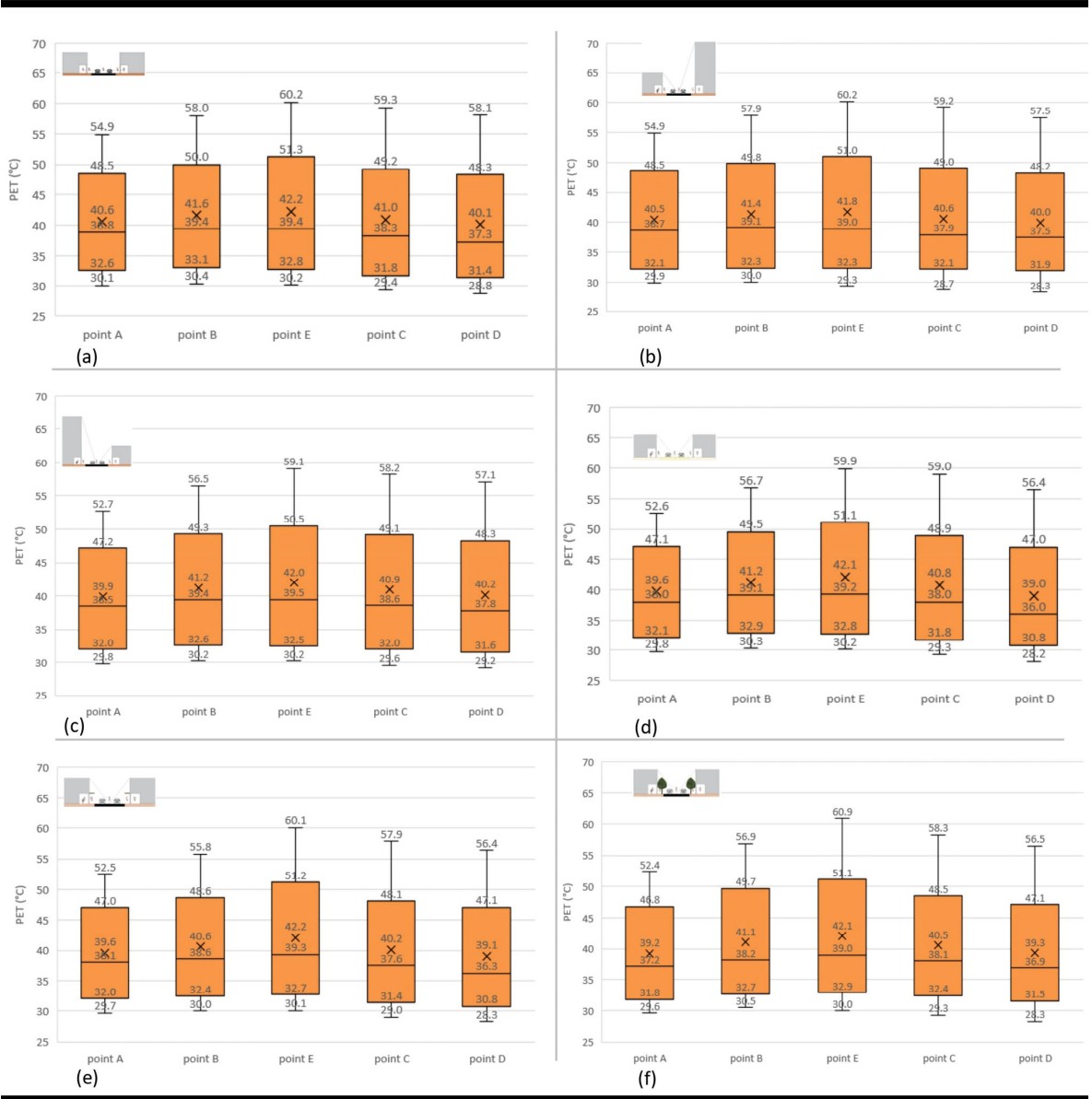

**Figure 10.** Box plots of the hourly PET values during 24 h on 08/08/2021 in points A, B, C, D, and E for (**a**) reference scenario, (**b**) scenario 1, (**c**) scenario 2, (**d**) scenario 3, (**e**) scenario 4, (**f**) scenario 5.

Overall, in the scenarios, SC4 achieved the optimal values in reducing PET but did not influence the middle-of-the-street point. The reduction reached 2.4 °C in point A, 2.2 °C in point B, 1.4 °C in point C, and 1.7 °C in point D. In this regard, SC4 presents the optimal reduction rate over 8 h compared to the previous scenarios. This result illustrates that directly implemented horizontal shading canopies above the pedestrian zone have a significant effective impact on PET values.

Scenario 5 (SC5) led to a significant reduction in PET index at points A and D (below the arcades), reaching 2.5 °C in point A and 1.6 °C in point D. However, compared to SC4 (horizontal shading), it did not yield sufficient results with regard to thermal mitigation at points B and C (directly under the tree canopy), with the maximum reduction equal to 1.1 °C in point B and 1.0 °C in point C. At point E, the PET values increased, reaching a maximum of +0.7 °C. This result can be explained by the relationship between the tree

canopy's shape and its leaves' orientation regarding solar radiation. The reflected short-wave radiation varied in different directions, which increased the Tmrt values in the middle of the street. It causes possible access to solar radiation during peak times, unlike the horizontal surface where the reflection occurs on a single side across the entire area.

In general, across all scenarios, point A shows favorable conditions in terms of OTC on the boulevard during the 12 hottest hours of the day. As shown in Figure 10, the distance between the median and maximum value lines was consistently the shortest at point A throughout all scenarios.

### 3.3. Spatiotemporal Distribution of PET

Figure 11 visualizes the spatial distributions of PET index levels within the different scenarios during the period between 07:00 and 22:00 on the second day of the study. The chosen hours represent the period of pedestrian high activity upon the boulevard. Using this representation method, we can assess the effect of different scenarios on the PET index over time at various points within the boulevard at the pedestrians' level.

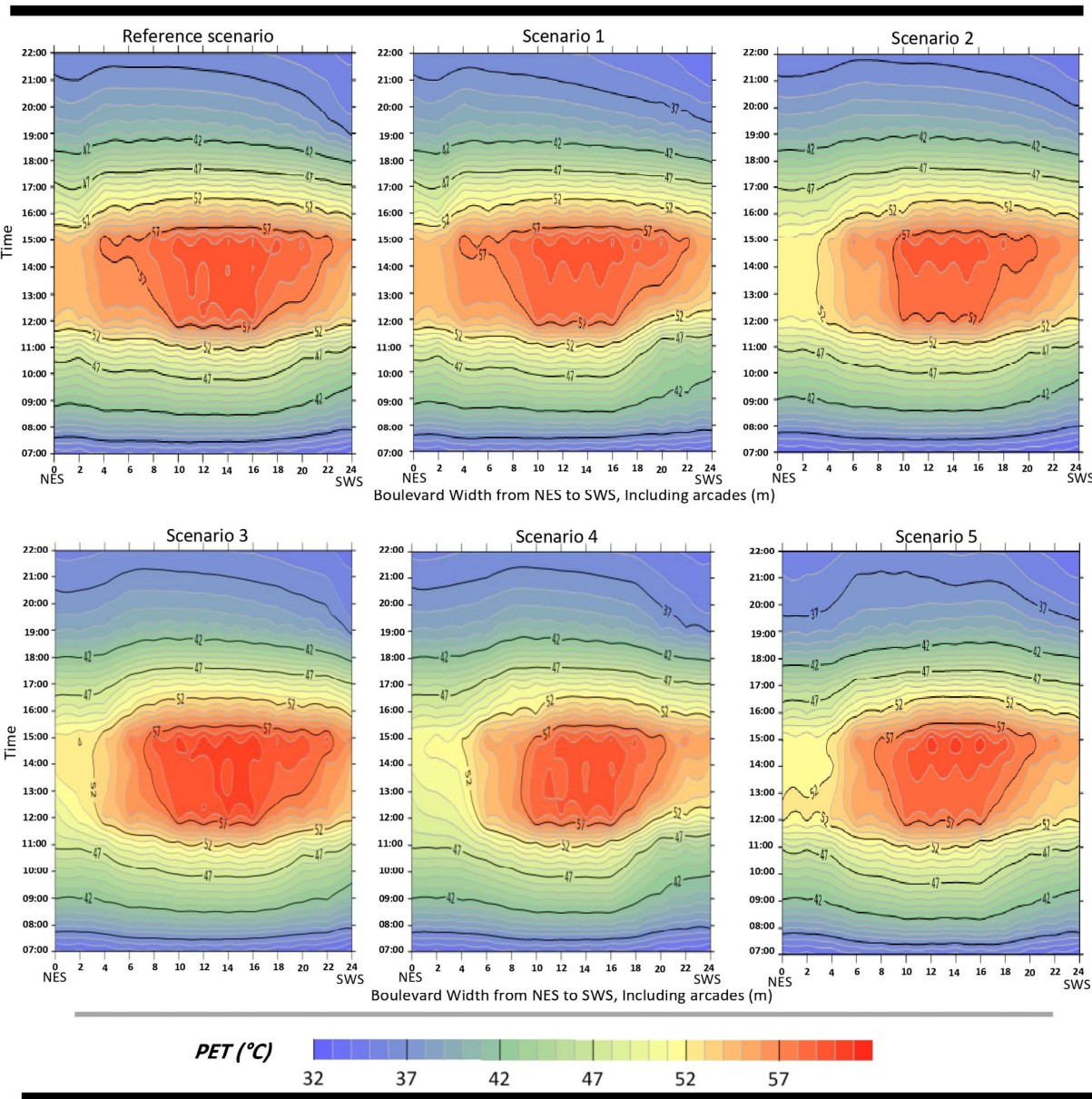

**Figure 11.** Spatiotemporal distribution of PET during the period between 07:00 and 22:00 on 8 August 2021 for different scenarios.

The results indicate that the period of extreme heat stress (PET > 42 °C) continues for 9 successive hours (approximately from 09:00 to 18:00) with a half-hour difference between both sides of the street. In this regard, it should be noted that all the scenarios can only reduce this duration by a few minutes. The most noticeable impact of the scenarios compared to the baseline scenario occurs during peak hours on the southwest orientation, where scenarios SC2, SC3, and SC4 can cool down the PET index levels below 52 °C for 3 h (approximately from 12:00 to 15:00) within 4 m of the northeast facade. Scenario SC5 also shows the same results but only for 2 h (from 13:00 to 15:00). The scenarios SC2, SC3, SC4, and SC5 can also reduce PET index levels below 57 °C within approximately 10 m from the northeast facade and within about 4 m from the southwest facade. A remarkable result for scenario SC1 was observed in the slight delay in the transition of PET from 42 to 47 °C, occurring approximately 0.5 h later, six meters from the southwest facade. The same result was achieved by scenario SC4.

In general, few other results were observed, such as a slight delay in the morning increase in PET or a slight acceleration in the evening decrease in PET, throughout all scenarios. On the other hand, the most significant impact was observed during the peak hours.

### *3.4. Findings and Recommendations*

Table 5 provides an overview of the main findings, emphasizing their agreements and divergences with previous studies. Additionally, it outlines suitable local design strategies based on the simulation outputs and PET index assessment.

This study aims to provide effective guidelines regarding the microscale design of wide streets in response to extreme weather conditions during the summer season in arid and hot areas. Based on modeling and simulation, it was possible to identify crucial urban recommendations that can be applied in the early urban design stages within similar environmental conditions.

Wide canyons receive high global solar radiation [23,110,111], considered a critical factor for the increase in human outdoor discomfort during summer conditions. The adaptation of the urban microscale at the pedestrian level to climate patterns can mitigate these conditions [112]. Therefore, the initial focus should be on utilizing the buildings as key elements to provide more shading within the outdoor spaces [44]. As shown in all the modeled scenarios, the sub-lateral areas throughout the canyon do not have similar thermal conditions [18,21,23]. Therefore, designing sidewalks in the usual symmetrical shapes is not recommended. The sidewalk's width should be evaluated by a detailed study which can take in several main criteria: (i) the shading potential on both sides and (ii) the duration and timing of intensive street activities. Thus, these criteria allow for maximizing the sidewalk width on the side that offers the best conditions.

Shading is one of the most efficient strategies for mitigating outdoor heat stress, especially when the implementation of shading canopies is on a limited scale [101,112]. However, it is crucial to ensure that the shading canopies maintain the level of openness for wide streets [44]. In this context, three shading strategies evaluated in this study can be discussed:

- Arcades on both sides of 'Emir Abdelkader Boulevard' provided optimal conditions throughout the daytime hours compared to adjacent areas. During peak hours, the difference in PET values between the boulevard's middle area and the SWS and NES in SC0, reached 5.3 °C and 2.1 °C, respectively. This design element was investigated in two previous studies and provided effective results in enhancing summer human-thermal conditions [37,86]. Unlike urban shading canopies, arcades benefit from the building structure as additional vertical protection from direct solar radiation and maintain lower temperatures within themselves during long daytime hours. It should be noted that the effectiveness of arcades depends on their width. Therefore, the addition of horizontal shading in SC4 (with an enlargement of arcade width from 4 m to 6 m) improved PET reduction within 2 m from the façade's edge by up to

2.4 °C. Accordingly, this strategy is a valuable tool for urban designers to enhance the pedestrian OTC, although it is essential to consider the required visibility and openness of the street. The width of arcades should not impart a feeling of enclosure.

**Table 5.** Study findings regarding mitigation strategies and design preferences.

| Strategy | Study Findings and Design Preferences |
|---|---|
| Asymmetry | • Similarly to previous studies [20,21,37,44], it has been confirmed that the strategy's effectiveness depends on the position of the highest building in relation to solar azimuth and shading potential throughout the daytime.<br>• For NW-SE street:<br> - Placing the highest building on NES can lead to optimizing PET values up to 2.0 °C during daytime hours (09:00–12:00). The greatest effect was recorded on N-E pedestrian areas.<br> - Placing the highest building on SWS can lead to optimizing PET values by up to 3.1 °C during peak hours (12:00–17:00). The greatest effect is recorded on S-W pedestrian areas.<br><br>Design preference:<br>According to Qaid and Ossen (2015), an asymmetrical profile obstructing the western sun in the latter half of the day is more desirable than the opposite profile obstructing the eastern sun [44]. Therefore, it is recommended to have an asymmetrical profile with an SWS > NES orientation. |
| High-albedo | • Regarding the OTC, the strategy's effectiveness depends on the quantity of the reflected solar radiation received from the high-albedo surfaces. This relationship is also discussed by Aboelata (2021), Rosso et al. (2018), and Santos et al. (2018) [18,32,63].<br>• Application of cool paving in shaded areas (especially below arcades) has led to optimizing the PET values by up to 2.3 °C. Several researchers have also highlighted the necessity of coupling cool paving surfaces with shading strategies to enhance pedestrian OTC [32,113,114].<br>• The implementation of cool paving in unshaded areas did not yield any significant results (maximum improvement of 0.3 °C). Our findings differ from studies that indicated increased thermal stress for pedestrians when cool paving is applied [115–117]. Nevertheless, Rosso et al. (2016) have highlighted some advantages obtained from high-albedo gravel over asphalt on the OTC [118].<br><br>Design preference:<br>Implementing cool paving on pedestrian areas under and outside arcades and incorporating shading canopies for the areas outside arcades, while excluding the use of cool paving on the road. |
| Shading | • Installing shading canopies directly over pedestrian areas is the most effective strategy for reducing heat stress during the summer season. Thus, many shading strategies have been recommended in several studies [21,24,39,101,110], whereas their effectiveness depends on the shape and the position of the canopy within the built environment.<br>• In the current study, horizontal shading supported by the building structure reduced the PET values by up to 2.4 °C in pedestrian areas. Shading strategies supported by buildings have not been previously studied. This result requires further examination.<br>• The linear arrangement of trees along building facades reduced the PET values in peak hours by up to 2.5 °C in pedestrian areas. However, it increased PET values in the middle of the street by up to 0.7 °C:<br> - Several studies have raised the positive impact of trees on OTC at the pedestrian level [48,51,55,119], which confirms that their effect is significantly perceived under the tree canopy during daytime hours [120–122]. Furthermore, it was confirmed that the trees' layout influences the spatial distribution of thermal physiological parameters [50,52,87,123].<br> - The study of Nouri and Costa (2017) pointed out that when trees are arranged linearly, the shades do not overlap due to their spacing arrangement; this configuration leads to high radiation levels occurring between these shaded areas [110]. Further research is required to fully understand the impact of parallel linear layouts on central areas between rows.<br><br>Design preference:<br>The implementation of horizontal shading on NES and tree shading on SWS (based on the best results from the modeled scenarios). |

• The horizontal shading implemented in the study can be considered as (i) an additional expansion for arcades and (ii) an alternative strategy for arcades in their reconstruction (in urban rehabilitation operations). Considering the summer and winter requirements,

horizontal shading can be used as a temporary solution for summer conditions. In this regard, the movable vertical screen developed by Swaid and Hoffman (1990) can be simulated [46]. A movable horizontal canopy can be installed as an extension of arcades during the summer and adjusted as needed in other seasons. It crucial to emphasize that the results obtained regarding horizontal shading depend on the material and thickness of the used building material surface [18]. This study specifically utilized a concrete surface with a thickness of 0.2 m.

- Street trees have led to a significant reduction in Ta levels during the daytime hours, emphasizing the importance of greenery in mitigating urban heat island (UHI) effects [14,53,57–59,123]. Furthermore, comparing urban design elements at the pedestrian level, specifically outside the arcades (directly below the trees' canopies and the horizontal shading surfaces), showed that horizontal shading achieved nearly optimal results of OTC. This result can be explained by (i) the flat concrete surface's ability to reflect a higher amount of short-wave radiation and (ii) the density of 'Ficus Retusa' trees used on both sides, which traps a higher amount of long-wave radiation emitted from shaded ground surfaces [101]. Using trees on one street side can provide sufficient space for the dispersion of long-wave radiation throughout the pedestrian area. Overall, the results regarding the effect of tree shading at the pedestrian level indicate that a limited-scale assessment is necessary to make optimized decisions regarding the position, arrangement, and types of planted trees [53–56,112].

Asymmetric streets represent an alternative strategy that maintains the required openness of main streets, while approximating the advantages of deep canyons on thermal conditions. The study of Qaid and Ossen (2015) shows a geometrical improvement over symmetrical deep canyons (that trap heat at night and impede airflow). High asymmetrical street buildings can enhance air circulation, disperse trapped heat, and provide large areas of street shading [44,124]. However, opposite asymmetric profiles show a remarkable variation in the mitigation levels between morning and evening periods. Therefore, we suggest conducting a preliminary field investigation to determine peak activity periods and adapt the suitable option for citizens' needs.

Regarding city cooling strategies, it is preferable to have an asymmetrical profile that can block the western sun during the latter half of the day, as opposed to obstructing the eastern sun [37,44]. Studies on asymmetric street orientations in hot and humid climates have confirmed their effectiveness in achieving a balance between summer and winter requirements, particularly with streets oriented between NW-SE and NE-SW [20,21]. For arid climates, further investigation is required.

Regarding the orientation of the studied boulevard, there is an agreement that placing the highest buildings on the SWS can provide better thermal conditions compared to the opposite profile [20,21,37,44].

Considering OTC at the pedestrian level, an essential caution must be exercised when implementing cool pavement strategies [18,22]. Cool paving can strongly increase the radiative exchange between pedestrians and the surroundings, resulting in elevated heat stress [30]. Based on the study results, it is evident that using cool paving in shaded areas can: (i) reduce Ta and (ii) decrease Tmrt values, thereby reducing heat stress in shaded areas. Similar findings have been observed in some previous studies [32,113,114]. Hence, we recommend evaluating the sensitivity to solar exposure by controlling the sky view factor intervals before choosing such pavements in the urban design process [18].

As a result, using cool paving on pedestrian areas below arcades decreased PET values on both sides during all daytime hours, whereas its use on the road resulted in a negligible result (a significant part of the road surface is exposed to solar radiation during daytime hours). In this regard, previous studies have shown different results concerning the use of cool pavement. Some studies have expressed worries about worsening thermal conditions when cool pavements were used [115–117], while others indicated the opposite [118]. However, using cool pavement in unshaded areas is not recommended due to the increased glare and negative impact on visual comfort [18,22,118].

## 4. Limitations of the Study

In the present study, we were unable to confirm the precision of Envi-met's outputs as they pertain to the evaluation of Tmrt. This limitation was due to the lack of tools for measuring global radiation, which is necessary for Tmrt calculation. Nevertheless, most studies rely solely on Ta to determine the program's validity to simulate different models under hot summer conditions [30]. Some research has implemented Tmrt to evaluate ENVI-met's performance, noting that the program provides a very reasonable approximation of this parameter at the pedestrian level [24,115,117]. Otherwise, several studies have demonstrated that the model tends to overestimate daytime Tmrt values but accurately reproduces the daily maximum values [30,95,97]. Consequently, we focused on examining our findings during the peak periods. It should be noted that the simulation time covered 7 and 8 August 2021 and was not a continuous assessment during a long-term period.

The fixed width and orientation of the studied street constrain the results. Any change in the street's orientation would lead to significantly different outcomes [20,23,36]. However, this study was conducted on a wide street with a low aspect ratio and focused on the pedestrian areas adjacent to the street boundaries. Many studies have shown that wide canyons (H/W < 1) yield highly reliable results in terms of microclimate along their boundaries [23,125,126]. Therefore, it can be trusted that our findings can be applied to northwest–southeast main streets in hot and arid climates.

This research was conducted to examine the effects of a hot desert climate on pedestrian thermal behavior within a wide street under summer conditions. The designed models were simulated using field measurements during an extreme heatwave, resulting in high PET index levels. Otherwise, it is worth noting that using measurements from representative summer days outside of the heatwave period could potentially affect the study's outcomes.

Despite the study's limitations, the research methodology can be replicable for an analysis in any region and different climate zones, which is strongly helpful for promoting climate-responsive streets that enhance citizens' quality of life.

## 5. Conclusions

Applying suitable urban strategies in outdoor spaces can greatly mitigate heat stress and promote walkability and cities' livability. This study investigated the effects of multiple criteria and their placement, such as an asymmetrical profile, high albedo, and shading, to mitigate the summer heat stress throughout the main streets of the city of Biskra in arid areas, serving as guidelines for similar climatic zones. Using a microscale approach, the study examined thermal comfort conditions in pedestrian areas within 'Emir Abdelkader Boulevard' to compare the potential solutions and to determine their optimal or near-optimal design configurations. Based on the study's main findings, several design guidelines and recommendations can be inferred, helping stakeholders and urban planners in various strategies for an adapted and sustainable urban design:

- Asymmetrical streets provide an alternative flexible geometry that maintains street openness and ensures more adaptability in modifying pedestrian thermal conditions. Opposite asymmetric profiles show a disparity in improving PET values between morning and evening hours. It is recommended to arrange the evening improvement option for city cooling strategies.
- Using high-albedo paving coupled with shading strategies is recommendable. High-albedo ground surfaces contribute to enhancing PET, which is initially improved under shading canopies. Using cool pavements in areas that lack shade had no significant impact on pedestrian thermal comfort. Therefore, several studies reported its negative effect on thermal and visual comfort. A control of the sky view factor is necessary when considering such materials.
- Adopting localized shading canopies is one of the most effective strategies for mitigating outdoor heat stress. Within the design options considered, each option offers some characteristics that should be taken into consideration:

- Arcades provide a crucial shading strategy, offering a mix of horizontal and vertical shading at the pedestrian level. Sub-areas below arcades on both sides offer better thermal conditions on the street throughout the day. Arcades' effectiveness depends on their depth. However, excessive depth should be avoided to prevent a sense of enclosure.
- Building-mounted horizontal shading provides a good alternative to arcade canopies in urban rehabilitation processes. It can also serve as an additional depth for them within specific seasonal settings. The effectiveness of this strategy depends on the canopy material, which may exceed the efficiency of tree shading.
- Single-sided tree planting is preferable for street design. When using trees (such as "Ficus Retusa" with large canopies) on both sides, the higher radiation levels occurring between tree shadows in the center of the street should be considered, as it can reduce the climatic performance of trees at the pedestrian level.

The obtained results made it possible to implement a reference design model according to the characteristics outlined in Table 6, serving as a recommendation for urban planners and landscape architects. The proposed model combines the best-performing features and configurations.

**Table 6.** Design preferences for NW-SE main streets in hot and arid climates.

| Strategy | Arcades | | Asymmetric Profile | | High Albedo | | Horizontal Shading | | Shading by Trees | |
|---|---|---|---|---|---|---|---|---|---|---|
| Position | SWS | NES | SWS > NES | NES > SWS | sidewalk | road | SWS | NES | SWS | NES |
| Preference choice | ✓ | ✓ | ✓ | ✕ | ✓ | ✕ | ✕ | ✓ | ✓ | ✕ |

Overall, the thermal conditions within the selected analysis days were far from the limits of neutral thermal sensation. For a better understanding of how different strategies effectively reduce heat stress and improve the walkability of a city, it would be necessary to conduct continuous and long-term studies. Additionally, the evaluation of the proposed scenarios was limited to their effectiveness as mitigation measures, with no consideration for their impact on the thermal conditions of an entire urban fabric. Our perspective is that, by utilizing a set of measurements to assess the urban microclimate conditions facing climate change, we can establish a reliable and practical approach.

**Author Contributions:** Conceptualization, H.N., M.E.M., S.B. and W.A.M.; methodology, H.N., M.E.M., S.B., W.A.M. and A.A.; software, H.N. and M.E.M.; validation, H.N. and A.A.; formal analysis, H.N.; investigation, H.N.; resources, H.N.; data curation, H.N. and M.E.M.; writing—original draft preparation, H.N. and M.E.M.; writing—review and editing, H.N., M.E.M., W.A.M. and A.A; visualization, H.N.; supervision, M.E.M.; project administration, M.E.M.; funding acquisition, H.N. and W.A.M. All authors have read and agreed to the published version of the manuscript.

**Funding:** This research received no external funding.

**Institutional Review Board Statement:** Not applicable.

**Informed Consent Statement:** Not applicable.

**Data Availability Statement:** Data used in this article can be obtained on request from the first author.

**Acknowledgments:** We would like to acknowledge the LACOMOFA Laboratory, University of Biskra, for the use of the monitoring equipment in this research and for valuable support during the experiments and data analysis. The authors would also like to thank the University of Biskra, Algeria, for their assistance in administrative procedures.

**Conflicts of Interest:** The authors declare no conflict of interest.

## Nomenclature

The following abbreviations are used in this paper:

| | |
|---|---|
| MBE | Mean Bias Error |
| N-E | Northeast |
| NES | Northeast Side |
| OTC | Outdoor Thermal Comfort |
| PET | Physiological Equivalent Temperature |
| RH | Relative Humidity |
| RMSE | Root Mean Square Error |
| R² | Coefficient of determination |
| S-W | Southwest |
| SWS | Southwest Side |
| Ta | Air temperature |
| Tmrt | Mean radiant temperature |
| Va | Air velocity |
| UHI | Urban Heat Island |
| CFD | Computational Fluid Dynamics |

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
