# Peer review of "Effect of Street Asymmetry, Albedo, and Shading on Pedestrian Outdoor Thermal Comfort in Hot Desert Climates"

_sustainability, doi:10.3390/su16031291_

Round 1

Reviewer 1 Report

Comments and Suggestions for Authors

Dear Author,

Some minor comments should be addressed before considering the manuscript for publication.

1- The text has some spelling errors; please check the entire text for spelling errors.

2- Please ensure the nomenclature list includes all acronyms and symbols throughout the manuscript.

3- The Abstract does not comprehensively reflect the necessary points that should be stated in this section to represent the entire manuscript. It would be a good idea to revise the Abstract accordingly. Further quantitative and qualitative points should be included.

4- A description of the proposed model and analysis, presumptions, related equations, model accuracy, and operating conditions should be provided. What are the main advantages of the current model?

5- More specific aspects of the work can be stated in the Conclusions section.

6- Line 314: should be R2.

7- Lines 321 and 322: should be °C.

8- Line 381: should be W/m2.

With kind regards,

Comments on the Quality of English Language

 Minor editing of the English language is required.

Author Response

Thank you very much, Reviewer 1. We appreciate your time and effort to provide feedback. We did our best to address all your comments in the revised manuscript. We marked responses in blue and defined the changes in the revised document by indicating the location (lines and pages) of the changed text.

Comments 1: The text has some spelling errors; please check the entire text for spelling errors.

Thank you very much, Reviewer 1. We appreciate your comment. We did our best to correct all the spelling errors throughout the entire manuscript following your suggestion.

Comments 2: Please ensure the nomenclature list includes all acronyms and symbols throughout the manuscript.

Thank you, Reviewer 1 . We agree with your comment. We have already noticed the absence of one acronym in the nomenclature list. You can find the added acronym to the nomenclature list on page 27, in the revised version.

Comments 3: The Abstract does not comprehensively reflect the necessary points that should be stated in this section to represent the entire manuscript. It would be a good idea to revise the Abstract accordingly. Further quantitative and qualitative points should be included.

Thank you very much, Reviewer 1 . We agree with your comment. We have added more clarfications and details throughout the abstract section.

Comments 4: A description of the proposed model and analysis, presumptions, related equations, model accuracy, and operating conditions should be provided. What are the main advantages of the current model?

Thank you very much, Reviewer 1. We appreciate your comment. We confirm that all the description of the numerical model and its analysis, presumptions, related equations, model accuracy and operating conditions are described on Table 1, ENVI-met settings and details of input data, and ENVI-met validation sections, throughout pages 8,9, and 10 of the revised version.

Comments 5: More specific aspects of the work can be stated in the Conclusions section.

Thank you very much, Reviewer 1. We appreciate your comment. We have incorporated a paragraph discussing the utilization of shading canopies as one of the most effective strategies for mitigating outdoor thermal stress, in lines 784-786, page 26, in the revised version. All other aspects are already described throughout the conclusion section.

Comment 6: Line 314: should be R2.

Thank you, Reviewer 1. We agree with your comment. We have made correction replacing R2 to R² in line 338, page 10, in the revised version.

Comments 7: Lines 321 and 322: should be °C.

Thank you, Reviewer 1. We agree with your comment. We have made corrections on (C°) to (°C) in lines 349-350, page 11, in the revised version.

Comments 8: Line 381: should be W/m2.

Thank you, Reviewer 1. We agree with your comment. We have made correction on (W/m2) to (W/m²) in line 410, page 12, in the revised version.

Comments on the Quality of English Language

Minor editing of the English language is required.

Thank you very much, Reviewer 1. We confirm that we have done our best to edit the quality of English language with a native speaker.

Reviewer 2 Report

Comments and Suggestions for Authors

In this study, the effect of street asymmetry, albedo and shading on the thermal comfort for pedestrian was simulated and analyzed. The title well describes the content, and the manuscript is well structured. The results are sound and solid, and the topic fits the scope of the journal. Overall, I would like to congratulate the authors for completing such an interesting and valuable work.

However, there are some issues to be improved in my opinion. My comments are listed as follows. I hope the authors find them useful for improving paper quality, and I wish the authors every success in the future.

1. In the Abstract, the research gap or limitation of current studies can be briefly mentioned.

2. To address the rising importance of thermal comfort and livability for human in a developing society, the following paper is recommended to be supplemented (DOI: 10.1016/j.apenergy.2022.119853). To address the effort of researchers in reducing air pollution and greenhouse gas emission, the following papers are recommended to be supplemented (DOI: 10.1016/j.seta.2022.102496; 10.1016/j.applthermaleng.2023.121300).

3. The version of software used in the study should be referred.

4. The resolution of Fig.11 can be improved.

5. It is recommended to give some other exampled regions with hot and arid climate. In this study, only a street in Algeria is modeled, while lots of countries or regions have hot and arid climate around the world. Giving some similar examples will be helpful for promoting interests for broad readers and enhance the impact of the research.

6. The authors have frankly introduced the limitation of this study. Even though the results are only valid for hot and arid climates, it is believed that the methodology can be used for regions of other climates. Therefore, in the ‘Limitation’ section, a future study can be discussed, and the significance of this study can be re-emphasized. For example, ‘although the results in this study focus on hot and arid areas, the research methodology in this paper can be used for the analysis in any region, which is helpful for promoting the climate-responsive streets that enhance citizens quality of life.’

Comments on the Quality of English Language

N/A

Author Response

Thank you very much, Reviewer 2. We appreciate your time and effort to provide feedback. We did our best to address all your comments in the revised manuscript. We marked responses in blue and defined the changes in the revised document by indicating the location (lines and pages) of the changed text.

Comments 1: In the Abstract, the research gap or limitation of current studies can be briefly mentioned.

Thank you very much, Reviewer 2. We agree with your comment. We confirm that we added several details about the research gap and limitation of current studies throughout the abstract section between lines 19-21, page 01, on the revised version.

Comments 2: To address the rising importance of thermal comfort and livability for human in a developing society, the following paper is recommended to be supplemented (DOI: 10.1016/j.apenergy.2022.119853). To address the effort of researchers in reducing air pollution and greenhouse gas emission, the following papers are recommended to be supplemented (DOI: 10.1016/j.seta.2022.102496; 10.1016/j.applthermaleng.2023.121300).

Thank you very much, Reviewer 2. We appreciate your comment. We confirm that we have added the suggested papers about pollution and greenhouse gas emission throughout the manuscript.

Comments 3: The version of software used in the study should be referred.

Thank you very much, Reviewer 2. We agree with your comment. We confirm that we added the ENVI-met software version throughout the text in line 280, page 08, in the revised version.

Comments 4: The resolution of Fig.11 can be improved.

Thank you, Reviewer 2. We agree with your comment. We confirm that we made improvements on the resolution of Figure 11, with 300 dpi resolution.

Comments 5: It is recommended to give some other exampled regions with hot and arid climate. In this study, only a street in Algeria is modeled, while lots of countries or regions have hot and arid climate around the world. Giving some similar examples will be helpful for promoting interests for broad readers and enhance the impact of the research.

Thank you, Reviewer 2. We appreciate your comment. We confirm that we have extensively discussed various climate zones and context, specifically focusing on mitigating outdoor thermal comfort within urban streets. We kindly invite Reviewer 2 to check the following references, which have already been discussed and cited in the paper:

‎1.‎  Cárdenas-Jirón, L.A.; Graw, K.; Gangwisch, M.; Matzarakis, A. Influence of Street Configuration on ‎Human Thermal Comfort and Benefits for Climate-Sensitive Urban Planning in Santiago de Chile. ‎Urban Clim 2023, 47, doi:10.1016/j.uclim.2022.101361.‎

‎2.‎  Rodríguez-Algeciras, J.; Tablada, A.; Matzarakis, A. Effect of Asymmetrical Street Canyons on ‎Pedestrian Thermal Comfort in Warm-Humid Climate of Cuba. Theor Appl Climatol 2018, 133, 663–‎‎679, doi:10.1007/s00704-017-2204-8.‎

‎3.‎  Deng, J.Y.; Wong, N.H. Impact of Urban Canyon Geometries on Outdoor Thermal Comfort in Central ‎Business Districts. Sustain Cities Soc 2020, 53, 101966, doi:10.1016/j.scs.2019.101966‎

‎4.‎  Ketterer, C.; Matzarakis, A. Human-Biometeorological Assessment of Heat Stress Reduction by ‎Replanning Measures in Stuttgart, Germany. Landsc Urban Plan 2014, 122, 78–88, ‎doi:10.1016/j.landurbplan.2013.11.003.‎

‎5.‎  Rodríguez-Algeciras, J.; Tablada, A.; Nouri, A.S.; Matzarakis, A. Assessing the Influence of Street ‎Configurations on Human Thermal Conditions in Open Balconies in the Mediterranean Climate. Urban ‎Clim 2021, 40, doi:10.1016/j.uclim.2021.100975.‎

‎6.‎  Shashua‐Bar, L.; Tzamir, Y.; Hoffman, M.E. Thermal Effects of Building Geometry and Spacing on the ‎Urban Canopy Layer Microclimate in a Hot‐humid Climate in Summer. Int J Climatol 2004, 24, 1729–‎‎1742.‎

‎7.‎  Qaid, A.; Ossen, D.R. Effect of Asymmetrical Street Aspect Ratios on Microclimates in Hot, Humid ‎Regions. Int J Biometeorol 2015, 59, 657–677, doi:10.1007/s00484-014-0878-5.‎

‎8.‎  Shashua‐Bar, L.; Potchter, O.; Bitan, A.; Boltansky, D.; Yaakov, Y. Microclimate Modelling of Street Tree ‎Species Effects within the Varied Urban Morphology in the Mediterranean City of Tel Aviv, Israel. Int J ‎Climatol 2010, 30, 44–57.‎

‎9.‎  Rosso, F.; Golasi, I.; Castaldo, V.L.; Piselli, C.; Pisello, A.L.; Salata, F.; Ferrero, M.; Cotana, F.; de Lieto ‎Vollaro, A. On the Impact of Innovative Materials on Outdoor Thermal Comfort of Pedestrians in ‎Historical Urban Canyons. Renew Energy 2018, 118, 825–839.‎

Comment 6: The authors have frankly introduced the limitation of this study. Even though the results are only valid for hot and arid climates, it is believed that the methodology can be used for regions of other climates. Therefore, in the ‘Limitation’ section, a future study can be discussed, and the significance of this study can be re-emphasized. For example, ‘although the results in this study focus on hot and arid areas, the research methodology in this paper can be used for the analysis in any region, which is helpful for promoting the climate-responsive streets that enhance citizens quality of life.’

Thank you, Reviewer 2. We agree with your comment. We confirm that we have added several details as suggested regarding the replicability of the research methodology within different context, between lines 759-761, page 25, in the revised version.

Reviewer 3 Report

Comments and Suggestions for Authors

Dear Authors, 

The study investigates street asymmetry, albedo, horizontal shading, shading by trees, and arcades on pedestrian outdoor thermal comfort to improve urban walkability in Algeria. The well-structured paper provides an informative simulation experiment on a particular summer day, representing the summer season. 

1- The paper's length certainly affects the flow of ideas, especially the introduction. I recommend substantially shortening the section; maybe adding a sub-heading will be beneficial. Also, a conclusion paragraph summarizing the estate-of-the-art, including the rationale for focusing on these elements, was added. 

2- The abstract should be improved, following the same research structure. Also, it contains a lot of abbreviations that may affect readers' focus. 

3. Table 5.: the study findings and design preferences column need to be unified in three elements to enable apple-to-apple comparison; for example, High-albedo and Shading present temperature values, while Asymmetry provides different aspects. In the same context, the table indicated shading as the third design strategy (in line with the text); however, in conclusion (lines 754-77), it presented additional design variables. 

4. Confirming the hypothesis and addressing the two research questions need to be discussed in the conclusion. 

5. In the limitations section, please indicate the period of conducting the simulation (one day in August) that might affect the results. However, it has been mentioned in line 178. 

* Please unify the citation style, e.g., line 140.
* The units are missing, e.g., line 131, °C of Ta

Finally, I  believe that the manuscript was made with a lot of effort. Although the data availability statement is clearly stated, it is recommended that the supplementary files be shared to support the experiment and open science practices.

Thank you 

Author Response

Thank you very much, Reviewer 3. We appreciate your time and effort to provide feedback. We did our best to address all your comments in the revised manuscript. We marked responses in blue and defined the changes in the revised document by indicating the location (lines and pages) of the changed text.

Comments 1: The paper's length certainly affects the flow of ideas, especially the introduction. I recommend substantially shortening the section; maybe adding a sub-heading will be beneficial. Also, a conclusion paragraph summarizing the estate-of-the-art, including the rationale for focusing on these elements, was added.

Thank you very much, Reviewer 3. We agree with your comment. We did our best to shorten the paragraphs’ length especially the introduction section. Moreover, we have added sub-heading throughout the introduction, as indicated in lines 56 and 94, on page 02, in the revised version. 

Comments 2: The abstract should be improved, following the same research structure. Also, it contains a lot of abbreviations that may affect readers' focus.

Thank you, Reviewer 3. We agree with your comment. We have added more clarfications and details throughout the abstract section in the revised version. Additionally, we added explanaitions regarding the abbreviations.

Comments 3: Table 5.: the study findings and design preferences column need to be unified in three elements to enable apple-to-apple comparison; for example, High-albedo and Shading present temperature values, while Asymmetry provides different aspects. In the same context, the table indicated shading as the third design strategy (in line with the text); however, in conclusion (lines 754-77), it presented additional design variables.

Thank you, Reviewer 3. We appreciate your comment. We have addressed your suggestion by unifying the comparaison parameter (PET index values) between Asymmetry, High-albedo, and Shading within Table 05. Furthermore, We have incorporated a paragraph discussing the utilization of shading canopies as one of the most effective strategies for mitigating outdoor thermal stress, in lines 784-786, page 26, in the revised version.

Comments 4: Confirming the hypothesis and addressing the two research questions need to be discussed in the conclusion.

Thank you, Reviewer 3. We appreciate your comment. We would like to confirm that we have already addressed the hypothesis and the research questions of the study regarding the implementation of mitigation strategies and we have provided appropriate placement for these strategies within the conclusion section. Although, we confirm that we included a text regarding the shading canopies throughout the conclusion, between lines 784-786, page 26, in the revised version.

Comments 5: In the limitations section, please indicate the period of conducting the simulation (one day in August) that might affect the results. However, it has been mentioned in line 178.

Thank you, Reviewer 3. We agree with your comment. We have added the period of conducting the simulation within the lines 743-745, on page 25, in the revised version.

Comment 6: Please unify the citation style, e.g., line 140.

Thank you very much, Reviewer 3. We appreciate your comment. It is important to note that two citation styles were employed throughout the text. Certain sections featured the complete citation, including the authors' names and the publication year, while other citations were presented solely as numbers in order of appearance in the text. Unfortunately, unifying all citation styles in the entire manuscript is not feasible.

We recognize the need for consistency in citation style and have carefully considered the suggestion. However, due to various factors specific to each section of the manuscript, we have chosen to maintain the current approach of utilizing both citation styles.

Comment 7: The units are missing, e.g., line 131, °C of Ta

Thank you very much, Reviewer 3. We agree with your comment. We have added the missing units throughout the text, in line 146, page 3, in the revised version. We should indicate that units in this paragraph were expressed in Kelvin (K) following the information reference. We kindly invite Reviewer 3 to check reference :

  • Santamouris, M. (2014). Cooling the cities–a review of reflective and green roof mitigation technologies to fight heat island and improve comfort in urban environments. Solar energy, 103, 682-703.

Comment 8: Finally, I  believe that the manuscript was made with a lot of effort. Although the data availability statement is clearly stated, it is recommended that the supplementary files be shared to support the experiment and open science practices.

Thank you very much, Reviewer 3. We appreciate your comments. We would like to confirm that we did an additional study technical report, containing all the measurements and simulation datasets. The report will be made available on Open Access (ORBI_University of Liege) once the paper is published. It will serve as a supplementary document to support the paper's results.

Round 2

Reviewer 2 Report

Comments and Suggestions for Authors

The authors well addressed my comments on the original version, and this revised version has a significant improvement on the quality. Therefore, I suggest an 'accept' for publication.

Reviewer 3 Report

Comments and Suggestions for Authors

The paper can be accepted in its present form. 
Thank you